# A Systematic Review of UAV Applications for Mapping Neglected and Underutilised Crop Species' Spatial Distribution and Health

Mishkah Abrahams [1,*], Mbulisi Sibanda [1], Timothy Dube [2], Vimbayi G. P. Chimonyo [3,4] and Tafadzwanashe Mabhaudhi [3,5]

1 Discipline of Geography, Environmental Studies & Tourism, Faculty of Arts, University of the Western Cape, Private Bag X17, Bellville 7535, South Africa; msibanda@uwc.ac.za
2 Institute of Water Studies, Faculty of Science, University of the Western Cape, Private Bag X17, Bellville 7535, South Africa; tidube@uwc.ac.za
3 Centre for Transformative Agricultural and Food Systems, School of Agricultural, Earth & Environmental Sciences, University of KwaZulu-Natal, P/Bag X01, Pietermaritzburg 3209, South Africa; v.chimonyo@cgiar.org (V.G.P.C.); t.mabhaudhi@cgiar.org (T.M.)
4 International Maize and Wheat Improvement Center (CIMMYT)-Zimbabwe, Mt Pleasant, Harare P.O. Box MP 163, Zimbabwe
5 International Water Management Institute (IWMI-SA), Southern Africa Office, Pretoria 0184, South Africa
* Correspondence: 3852812@myuwc.ac.za

**Abstract:** Timely, accurate spatial information on the health of neglected and underutilised crop species (NUS) is critical for optimising their production and food and nutrition in developing countries. Unmanned aerial vehicles (UAVs) equipped with multispectral sensors have significantly advanced remote sensing, enabling the provision of near-real-time data for crop analysis at the plot level in small, fragmented croplands where NUS are often grown. The objective of this study was to systematically review the literature on the remote sensing (RS) of the spatial distribution and health of NUS, evaluating the progress, opportunities, challenges, and associated research gaps. This study systematically reviewed 171 peer-reviewed articles from Google Scholar, Scopus, and Web of Science using the PRISMA approach. The findings of this study showed that the United States ($n = 18$) and China ($n = 17$) were the primary study locations, with some contributions from the Global South, including southern Africa. The observed NUS crop attributes included crop yield, growth, leaf area index (LAI), above-ground biomass (AGB), and chlorophyll content. Only 29% of studies explored stomatal conductance and the spatial distribution of NUS. Twenty-one studies employed satellite-borne sensors, while only eighteen utilised UAV-borne sensors in conjunction with machine learning (ML), multivariate, and generic GIS classification techniques for mapping the spatial extent and health of NUS. The use of UAVs in mapping NUS is progressing slowly, particularly in the Global South, due to exorbitant purchasing and operational costs, as well as restrictive regulations. Subsequently, research efforts must be directed toward combining ML techniques and UAV-acquired data to monitor NUS' spatial distribution and health to provide necessary information for optimising food production in smallholder croplands in the Global South.

**Keywords:** crop health; drones; food security; NUS; precision agriculture; spatial distribution; stomatal conductance; UAV

## 1. Introduction

Inherent water scarcity, which is exacerbated by factors such as climate change, population expansion, and changes in land use [1–3], has intensified the pressure on the agricultural sector, particularly in ensuring long-term food supply for the expanding populations [4,5]. Most of the agricultural production in developing regions is derived from rainfed farms, which occupy 97% of croplands [6]. However, 80% of these croplands are

smallholder farms that contribute most of the food production in developing regions [1,2]. However, these croplands are marginal due to suppressing agronomic and climatological infrequencies. Therefore, climate variability and soil degradation have drastically reduced the production of staple cereal crops, such as maize, amplifying food and nutrition security issues in these regions. This has placed local food systems on the verge of catastrophe. Hence, establishing innovative methods to combat food and nutrition insecurity while optimising production is urgently needed.

A paradigm shift away from cultivating susceptible cereal crops towards diversifying climate-smart crops, such as underutilised crop species (NUS), is necessary [7]. NUS, also referred to as alternative, traditional, orphaned, or neglected crops, are adapted to flourish in fragile production systems where land degradation and drought are topical [3]. Examples include millet, quinoa, teff, Bambara ground nuts, sweet potato, and taro [3–5]. These crops are stress-tolerant, require fewer inputs, and are highly adaptable to a broad spectrum of ecological niches [8]. Including NUS in fragile production systems is one of many strategies for safeguarding the well-being of minority populations, meeting sustainable development goals (SDGs), leveraging indigenous knowledge systems, and facilitating traditional food and cultural heritages.

A major challenge to the mainstreaming of NUS is that basic information on genetic and phenotypic traits remains scanty and localised, and productivity is generally low. Generating spatially explicit information on NUS' biochemical and morphological characteristics to optimise their productivity could contribute to their development and promotion. In this regard, spatially explicit, high-throughput phenotyping technologies are required to augment the rapid advancements in phenotyping NUS. This will aid in maximising their productivity in fragmented smallholder croplands.

Generally, field surveys and other traditional methods have been widely used to measure the spatial extent, suitability, growth, and morphological attributes of NUS. However, such methods are time-consuming and expensive, making them unsuitable for continuous precision crop monitoring in areas with numerous crop varieties. Over the past few decades, satellite-based Earth-observation technologies, such as Landsat and MODIS, have become effective for monitoring plant growth and health changes [9]. Satellite-borne remote sensing technologies provide non-invasive, accurate, fast, and cost-effective data for estimating traits such as stomatal conductance and chlorophyll content as a proxy for crop productivity, hence their use [10]. However, the resolution of these freely available satellite sensors provides limited information for high-throughput phenotyping, particularly in the context of highly fragmented and diverse smallholder croplands [11]. In this regard, smallholder croplands require very high-spatial-resolution remote-sensing technologies that are affordable and highly efficient [11].

The advent of UAV-based phenotyping allows access to data with extremely high levels of detail and precision. This is ideal for the precise, consistent phenotyping of smallholder farms at the plot level [11]. However, the capacity of UAV-mounted sensors to differentiate crop types based on their spectral responses as a mechanism for plausible high-throughput field phenotyping is yet to be determined [12]. RS and ML techniques have recently greatly aided high-throughput phenotyping technologies [12]. For example Li, Burnside [6] combined three ML techniques, which included random forest (RFR), support vector regression (SVR), and artificial neural network (ANN), in combination with optimal VI's to predict the red-clover dry matter yields in various phenological growth periods.

Despite the usefulness of UAVs, their application in agriculture, rural development, and, more importantly, resource management remains limited [13]. Although some studies have attempted to assess the literature on the application of drone-acquired data [14,15], most of these studies did not systematically and quantitatively assess the literature on mapping the spatial distribution and health of NUS crops with a special interest in Global South trends. The objective of this study was to systematically review the literature on the application of UAV remotely sensed data for mapping the spatial distribution and

health of NUS, with a particular focus on sweet potatoes and taro. The aim was to gain an understanding of the progress, challenges, opportunities, and gaps in this field. Gaining insights into the mapping of NUS' spatial extent, morphological features, and biochemical traits through remotely sensed data acquired by UAVs could pave the way for enhancing food production in South Africa's smallholder croplands.

## 2. Materials and Methods

### 2.1. Literature Search

This review followed the Preferred Reporting Items for Systematic Reviews and Meta-Analyses (PRISMA) checklist and guidelines for gathering and analysing the literature. In the initial literature search phase, keywords, terms, and phrases for searching the literature were generated from other literature reviews on NUS [16–18]. The methodology followed in this study was adopted from the literature [8–10,19]. The following keywords and their variants were used in this study to search for relevant literature: "neglected and underutilised crop species", "orphan crops", "traditional crops", "unmanned aerial vehicle(s)", "drone(s)", "remote sensing", "GIS", "crop health", "stomatal conductance" and "leaf area index", and "chlorophyll". Furthermore, the following PRISMA statement and its variants were used to search for research pertaining taro and sweet potatoes: "Taro", "sweet potato", "unmanned aerial vehicle(s)", "drone(s)", "remote sensing", "GIS", "crop health", "stomatal conductance" and "leaf area index", and "chlorophyll".

The SCOPUS, Web of Science, and Google Scholar databases were utilised to collect literature using the established key search terms. The PRISMA statement served as the framework for the literature search procedure. This search was not restricted in terms of time. The Google Scholar, Scopus, and Web of Science literature searches yielded 109, 1036, and 90 articles, respectively (*n* = 1235). In preparation for screening, all obtained material was organised in EndNote. The screening procedure considered in this study followed the PRISMA procedure, reported as a flowchart (Figure 1). For an article to be considered in the meta-analysis, it had to meet the following criteria;

(a) The study focuses on NUS crops, traditional, or orphaned crops, and no other vegetation types (e.g., forests or shrubs) were included, since they denoted different ecosystems;
(b) The study focuses on NUS productivity (i.e., LAI, chlorophyll, or stomatal conductance) or spatial distribution;
(c) The study was based on UAV or drone remotely sensed data, GIS, or remote-sensing techniques in NUS crop productivity and health mapping;
(d) The article was published in an accredited journal;
(e) The article was written in English;
(f) The article was accessible.

After eliminating all duplicates (*n* = 650), 585 remained. In this case, literature not written in English was excluded from the analysis (*n* = 20). The next step was to assess whether the retrieved literature covered the context of mapping the spatial distribution or assessed productivity elements of neglected and underutilised crops based on remotely sensed data by examining the abstracts. After the title and abstract screening, 332 studies were excluded and 253 remained. Of the remaining articles, 82 were not accessible as full texts and were excluded, with 171 articles remaining. The full-length articles of the selected abstracts were then sought and downloaded as PDF documents. After the screening procedure, 171 articles were retained (Figure 1). Then, the bibliographic information, including author names, year of publication, title, journal name, issue, volume number, keywords, and abstracts, was exported from Endnote as a text file to Microsoft Excel. The Excel database was then used to extract and store qualitative and quantitative data from each article, as indicated in the proceeding phase.

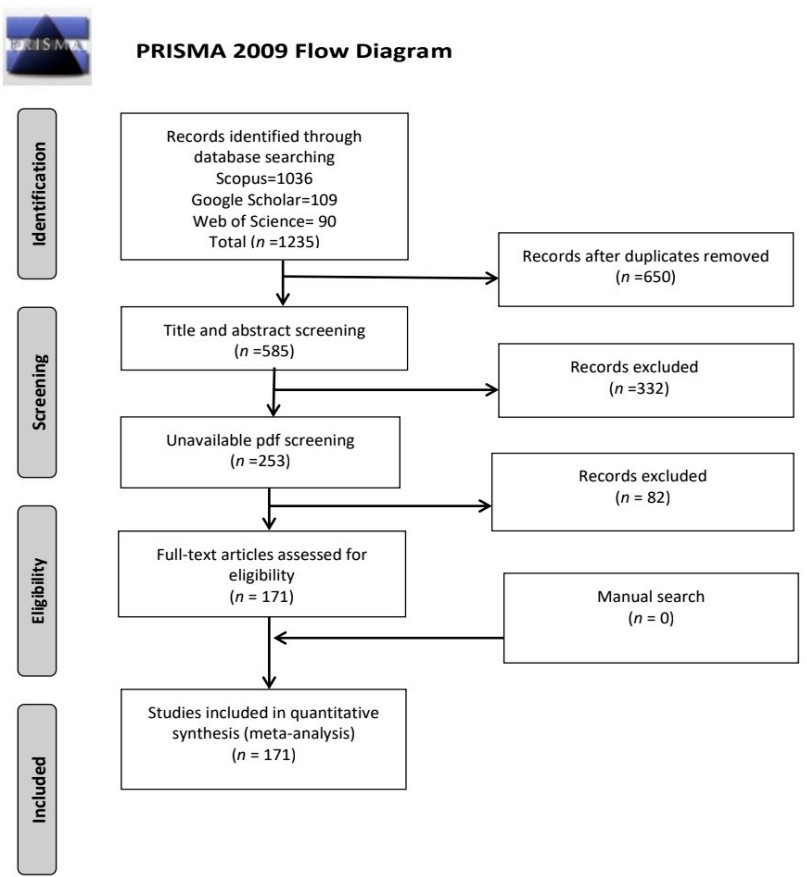

**Figure 1.** PRISMA flow diagram for selection of studies considered in the review.

### 2.2. Data Extraction

In the preceding phase, the extracted data in an Excel database were used to generate and extract information on progress, gaps, challenges, and opportunities of using UAV technologies to map the spatial distribution of different NUS and their health attributes. Specifically, information related to the study country, region, NUS type, the type of crop health attribute investigated, sensor type and platform type, vegetation indices, predictive or classification algorithms, and optimum spectral variables obtained, were all retrieved from the literature and documented in the spreadsheet. Some categorical variables were converted into numeric values to facilitate analysing and evaluating trends in the retrieved literature. During this step, relevant bibliometric information was also collected. As mentioned above, the qualitative and quantitative information extracted from each article were added to the Excel spreadsheet with the author names, region, year of publication, article title, journal name, and abstract, among the other bibliometric data gathered.

### 2.3. Data Analysis

The retrieved literature and extracted data were subjected to quantitative and qualitative analyses during this phase. Basic statistical frequencies were calculated for quantitative analysis [13]. In addition, exploratory trend analysis was carried out on the frequency of publications to assess the progress made in mapping NUS' spatial distribution and health attributes utilizing satellite and drone-borne sensors. A bibliometric analysis was also conducted to identify trends in co-occurring key terms from the retrieved literature. The trends were identified by quantitatively examining the occurrence and co-occurrence of key terms in the titles and abstracts using the VOS viewer software. Furthermore, the VOS viewer software used the titles and abstracts of all the retrieved literature (171 articles) and only those based on UAV-derived datasets (18 articles). This assisted in evaluating how concepts and topics evolved in mapping NUS using satellite-based remotely sensed data to

drone-acquired data. VOS viewer provides network visualisation of key terms in linked clusters. Creating a map in VOS viewer includes four steps, which are:

(1) Selecting a counting method (binary counting or full counting);
(2) Selecting a minimum number of occurrences for a term (calculating similarity index);
(3) Calculating the relevance score for the co-occurrence terms and displaying the most relevant items based on this score;
(4) Displaying a map based on the selected terms.

Considering that only the occurrence, co-occurrence of key terms, and frequency distributions were computed, bias assessment was not conducted. As aforementioned, the PRISMA checklist (http://www.prisma-statement.org/, accessed on 1 July 2022) was used as a guideline to avoid biased reporting. In this regard, no further robust bias statistical assessments were conducted, since only exploratory data analysis was conducted. The review was divided into two main sections to address the research objectives. The first section investigated recent advances in mapping NUS crops' spatial distribution and health using remotely sensed data. This section presented and discussed quantitative literature trends in analysing NUS's spatial distribution and health. Throughout this phase, the crop health attributes, Earth-observation sensors (cameras), sensor platforms, algorithms, and optimal spectral variables used by the community of practice in the retrieved literature were assessed and presented. The last phase discussed the challenges, gaps, and opportunities for knowledge generation in mapping NUS's spatial distribution and health using drone-derived remotely sensed data.

## 3. Results

Figure 2 shows the co-occurrence of topical concepts derived from titles and abstracts from (a) studies based on satellite-borne data and (b) drone-acquired data. Figure 2a illustrates seven topical clusters in dark blue, light blue, red, green, purple, yellow, and orange for mapping crop spatial extent and health status. The key terms from the "red" cluster were "agriculture", "spad value", "low cost", "remote sensing data", "hyperspectral data", "multispectral data", "VIS", "processing", "size", and "crop field", which directly imply the utility of "low-cost remote sensing" systems for mapping and monitoring NUS crop productivity (spad-value/chlorophyll) with remotely sensed data in smallholder crop fields (Figure 2a). The second-largest cluster linked to UAVs was in yellow and contained "growth", "variety", "trait", "detection", "plant-height", "UAVs", "drone", "dsm", "rgb", and "msi". This cluster articulates the use of drone remotely sensed ("UAVs") data in detecting and mapping relevant NUS phenotypical attributes ("growth", "variety", "trait", "detection", and "plant height"). The third cluster in dark blue included 'growth stage', "lai", "crop height", "agdw", "canopy nitrogen", "weight", "fusion", "plsr", and "rmse" as the key terms in order of importance (Figure 2). This cluster relates to the estimation of NUS crop productivity attributes ("lai", "crop height", "agdw", "canopy nitrogen", "weight", "lai", "crop height", "agdw", "canopy nitrogen", and "weight") using remotely sensed data and regression techniques. The fourth cluster in green contained "population", "climate change", "water stress", "food security", "African leafy vegetable", "suitable area", "moisture", "SSA" (Sub-Saharan Africa), and "validation", among others. This links to the food and nutrition security issues in SSA, which are highly impacted by climate variability, such that NUS crops ("African leafy vegetable") are the only suitable crops because of their drought tolerance. The fifth cluster, which is light blue, features co-occurring terms, such as "species", "reflectance", "density", "waveband", "nir", "classification", and "palmer amaranth". This cluster relates to the impact of species variability and plant or foliage density, which causes the spectral signatures of NUS crops to vary at different wavebands of the EM spectrum, facilitating optimal classification accuracies. The sixth cluster is characterised by the co-occurrence of terms such as "plant", "amaranth", "phenological", and "growth stage", which relates to the assessment of NUS crops' phenological characteristics.

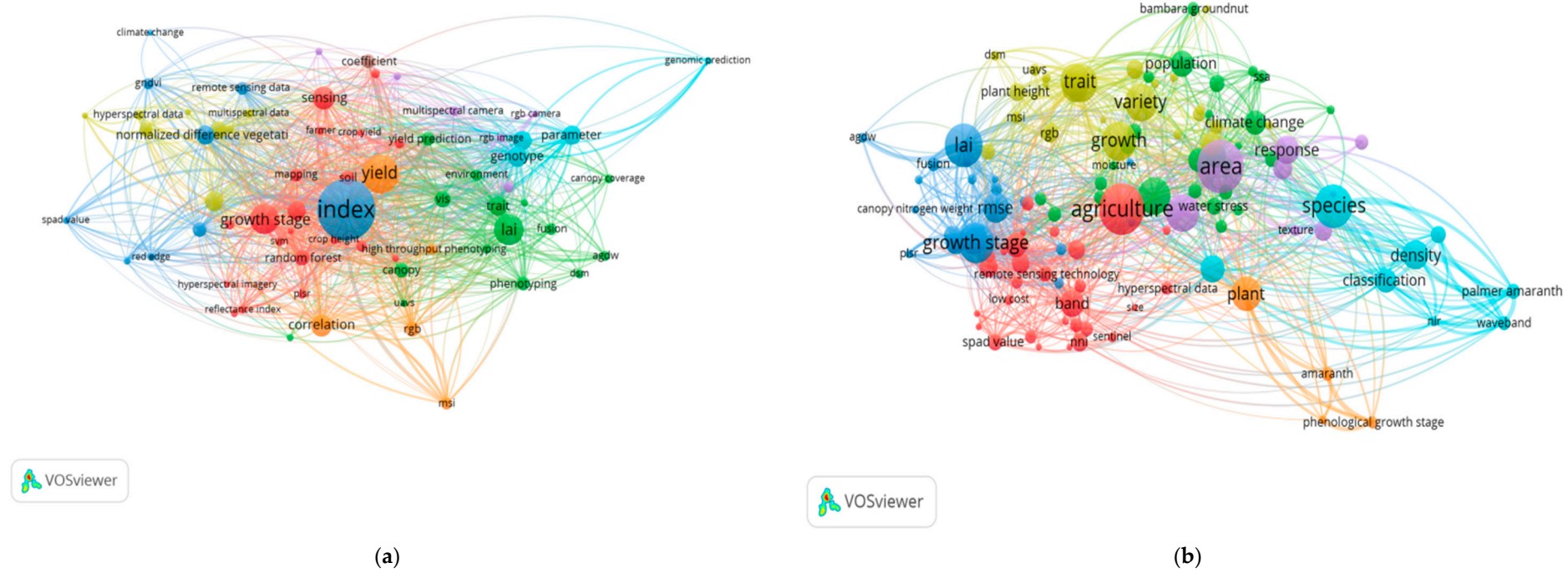

**Figure 2.** Topical concepts identified using a bibliometric analysis of titles and abstracts of articles that utilised (**a**) all remote sensing sensors and (**b**) only drone-acquired data in mapping the NUS. Various coloured lines establish connections between keywords that co-occurred within the same documents.

Figure 2b illustrates seven topical clusters in dark blue, light blue, red, green, purple, yellow, and orange, which were derived using abstracts and titles that utilised drone remotely sensed data in mapping NUS crop spatial extent and healthiness. The key terms from the red clusters were "mapping", "crop height", "growth stage", "yield" and, "crop yield", "reflectance index", "hyperspectral imagery", "random forest", and "svm". This cluster relates to using high-resolution remotely sensed data in mapping and monitoring NUS productivity elements, such as growth stage, using machine learning. The key terms in the green cluster are "phenotyping", "LAI", "canopy coverage", "trait", "agdw", "prediction", "uavs", "dsm", "fusion", "VIS", and "environment". This cluster relates to using UAV remotely sensed data that could be fused with other data for assessing the structural attributes of NUS crops (phenotyping). The third cluster in dark blue has the key terms of "remotely sensed data", "gndvi", "normalised difference vegetation", "red-edge", "spad value", and "climate change". This cluster relates to the utility of the widely used spectral variables ("gndvi", "normalised difference vegetation", and "red-edge") in monitoring the health of NUS crops. The fourth cluster is orange and has co-occurrence terms, such as "high throughput phenotyping", "rgb", "msi", "yield", and "correlation". This cluster can be attributed to applying the high-throughput phenotyping of NUS through RGB and multispectral spectrums in estimating crop yield. The fifth cluster in light blue had "genotype", "parameter", and "genomic parameters". This cluster suggests examining the genetic composition of NUS crops through the assessment of optimal parameters to assess and infer their crop health. The sixth cluster has "multispectral camera", "rgb camera", "rgb image". This cluster relates to the use of general colour imagery (in red, green, and blue spectrums) acquired using drones for mapping NUS. The last cluster has "hyperspectral data" and "multispectral data". This cluster relates to the utility of multispectral and hyperspectral data in mapping and monitoring NUS crop spatial distributions and health.

*3.1. Progress in Mapping the Spatial Distribution and Health Status of Neglected and Underutilised Crop Species*

Significant progress has been attained in detecting, mapping, and monitoring NUS crops' spatial distribution and health status using remotely sensed data (Figure 3). However, it should be noted that this progress relates to the collective of NUS, but not each individual species. The period between 2003 and 2013 is marked by a low frequency of published literature based on all Earth-observation sensors (Figure 3). Between 2014 and 2022, there was a rapid increase in articles that mapped the health and spatial distribution of NUS using all Earth-observation sensors. Again, from 2014 to 2022, there was a rapid growth in the literature that utilised UAV-acquired remotely sensed data to characterise NUS attributes. Despite substantial progress, only a few studies have demonstrated the effectiveness of remote sensing technologies in NUS crop classification, characterising their suitability ranges, or discriminating their varieties based on their phenotypic traits. Specifically, only 3% and 4% of the retrieved studies utilised drone and satellite-borne remotely sensed data in mapping the spatial distribution of NUS crops, respectively. However, most of the retrieved literature evaluated NUS phenological and phenotypic characteristics.

Regarding geographic distribution, the studies included in the meta-analysis were conducted in 47 countries. Most of the retrieved literature was conducted in Asia, America, and Africa. On a national scale, most of the studies were conducted in the United States of America ($n$ = 18) [11,12], followed by South Africa ($n$ = 18) [13,14] and China ($n$ = 17) [15,16], (Figure 4). Interestingly, most of the African studies were conducted in southeast Africa.

The most prevalent NUS in the retrieved literature included sweet potato (*Ipomoea batatas*), sorghum (*Sorghum bicolor*), amaranth (*Amaranthus cruentus*), cassava (*Manihot esculenta*), cowpea (*Vigna unguiculata*), millet (i.e., pearl (*Pennisetum glaucum L.R. Br.*), finger (*Eleusine coracana*), and proso millet (*Panicum milliaceum*)) (Figure 5a). UAV-based remotely sensed data was utilised in mapping the attributes of amaranth ($n$= 6) [17], Legume (*Fabaceae*) ($n$ = 2) [18], Sweet potato ($n$ = 2) [20], sorghum ($n$ = 2) [21], and bambara groundnut (*Vigna subterranea*) ($n$ = 2) [22] (Figure 5b). Most of the retrieved literature

remotely sensed various NUS using hyperspectral data, hence the decline in the frequency of NUS studies that utilised drone and satellite-acquired remotely sensed data (Figure 5).

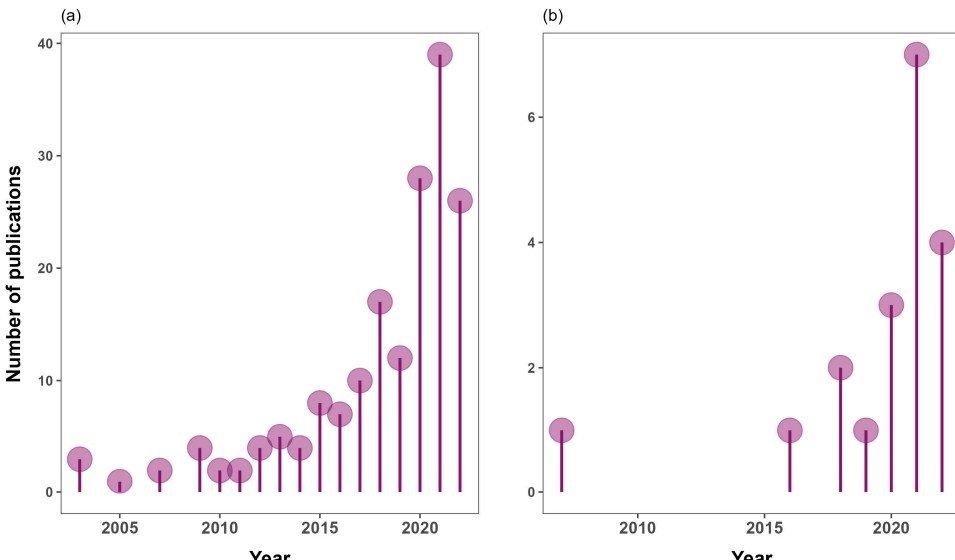

**Figure 3.** Frequency of published articles on remote sensing applications of NUS based on (**a**) all sensors and (**b**) drones.

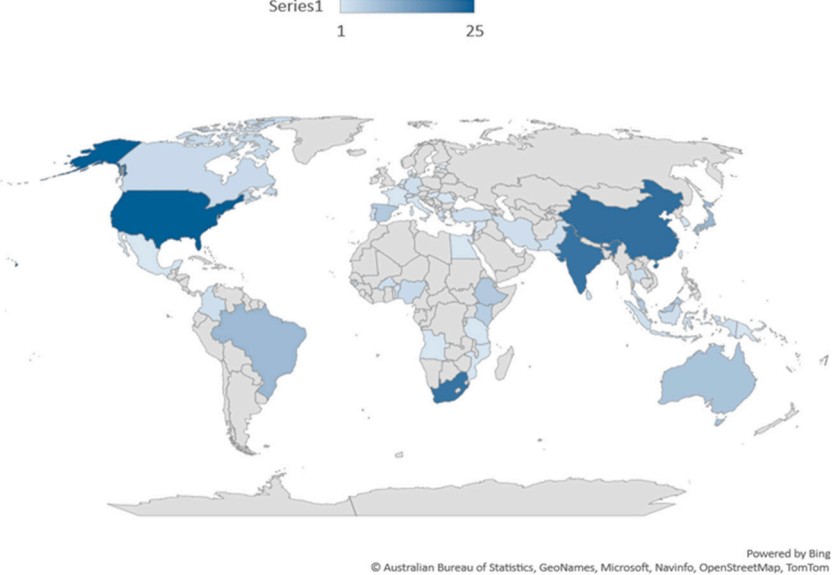

**Figure 4.** Spatial distribution of studies on remote sensing the attributes and spatial distribution of NUS.

Eight key broad research themes emerged from the reviewed literature on NUS. These include phenotyping, crop genetics, crop productivity, crop physiology, phenology, crop adaptation, classification, and land suitability (Figure 6). Most reviewed studies focused on quantifying NUS' physiological and phenological crop traits. A relatively small number of studies characterised crop classification and spatial extent. Five studies discriminated crops based on drone-acquired data, while six articles were based on satellite-borne remotely sensed data (Figure 6). The most extensively researched areas based on all satellite-borne sensors included phenology (*n* = 142), crop physiology (*n* = 110), and crop productivity (*n* = 104). When considering only the drone-borne sensors, 16 studies focused on NUS phenology. These studies mainly assessed external crop attributes, such as crop LAI, biomass, crop height, and crop yield periods across the growing season. For example,

Jewan, Pagay [23] monitored six distinct phenological stages of the Bambara groundnut, estimating its yield. Only nine studies assessed NUS crop productivity, while seven research studies examined NUS crop health and physiology. For instance, Lati, Avneri [24] assessed the utility of a UAV imaging platform coupled with an RGB sensor to monitor chickpea's physiological and morphological parameters, such as LAI, biomass, and yield, during irrigation periods.

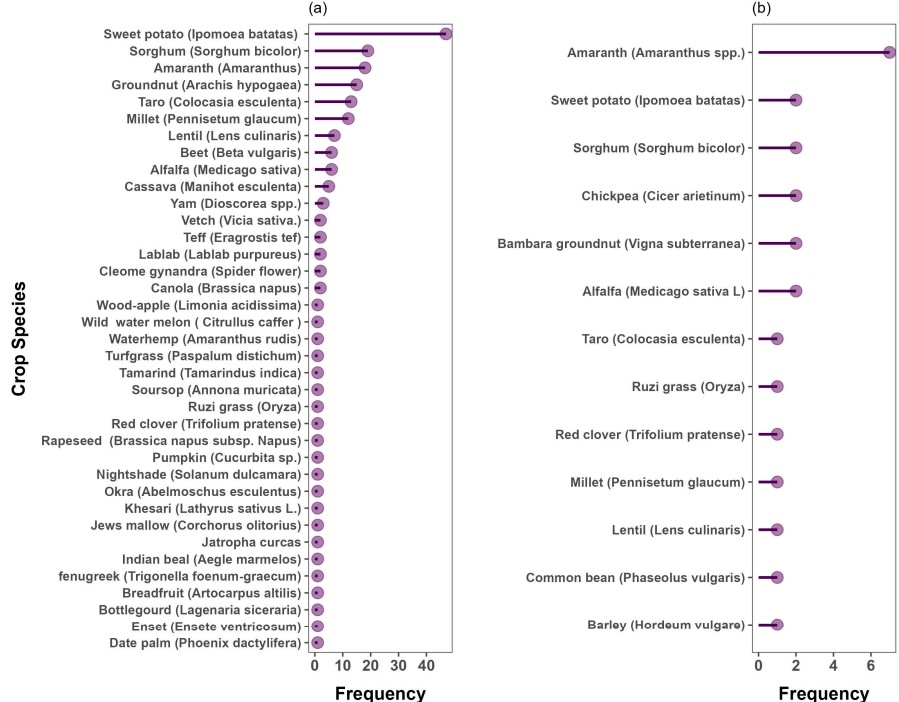

**Figure 5.** Frequency of NUS in the literature remotely sensed using (**a**) all various sensors and (**b**) exclusively drone-borne sensors.

Moreover, five studies focused on characterizing NUS crop spatial distribution using classification methods, and four concentrated on NUS's climatic suitability and adaptation. For example, Ramírez, Grüneberg [20] utilised vegetation and temperature indices to characterise various sweet potato genotypes based on productivity and resilience under drought treatments. Only five studies utilised crop phenotyping and breeding techniques. In many instances, crop phenotyping and genotyping included single-nucleotide polymorphism (SNP) markers. SNP markers provide a broad range of applications in various crops, including plant variety and cultivar characterization, quantitative trait loci (QTL) analysis, the production of a high-density genetic map, and genome-wide association analysis [25]. And lastly, four studies focused on land suitability.

Many of the retrieved studies mapped the productivity and water stress-related elements. Specifically, the most extensively researched NUS health attributes included crop yield, growth attributes, crop health, chlorophyll content, leaf water content, biomass, photosynthesis, LAI, stomatal conductance, canopy height, plant weight, leaf nitrogen, and canopy temperature (Figure 7a). The most researched NUS attributes in the context of UAV-based remote sensing were crop yield ($n = 15$), growth attributes ($n = 13$), biomass ($n = 11$), crop health ($n = 10$), chlorophyll content ($n = 9$), canopy cover ($n = 7$), plant/canopy height ($n = 7$), LAI ($n = 6$), leaf nitrogen ($n = 5$), leaf size attributes ($n = 5$), leaf water content ($n = 3$), and leaf temperature ($n = 3$). As aforementioned, there were very few studies that classified and characterised the spatial distribution of NUS (Figure 7b) [4].

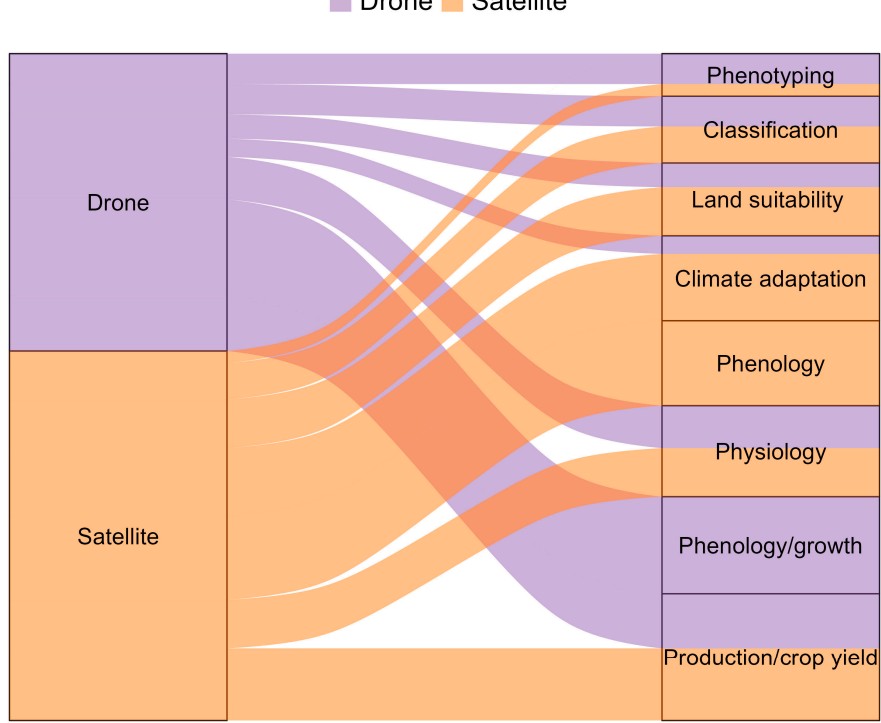

**Figure 6.** Frequency of articles that utilised remotely sensed data from drones and satellites to map NUS attributes. The thickness of the lines represents the frequency in the retrieved literature. (See Supplementary Materials spreadsheet 454 for frequency values).

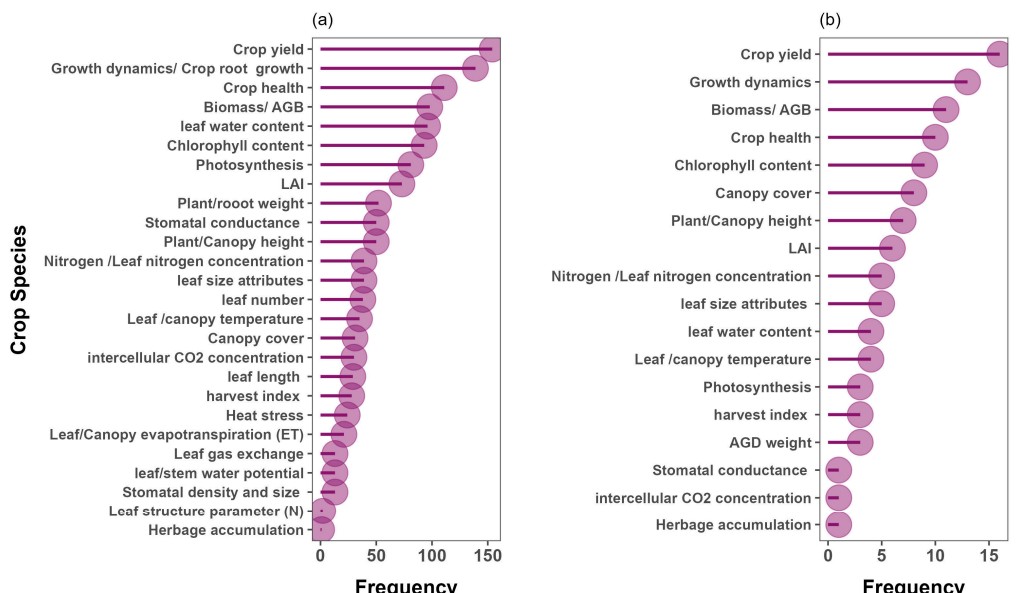

**Figure 7.** Frequency of studies that remotely sensed a specific crop attribute based on (**a**) all satellite and drone-borne sensors and (**b**) drone-borne sensors only.

### 3.2. Assessing Literature on Classification and Stomatal Conductance Estimation of Taro and Sweet Potato Crops

Based on the findings, 12 articles characterised the spatial distribution of various NUS (Figure 8). Sweet potato, lentil, and chickpea were the crops that received substantial attention in the literature (Figure 8). Overall, a limited number of studies have utilised UAV remotely sensed data to classify NUS. There is a gap in the research focusing on UAV classifications from 2008 to 2019. Furthermore, 2020 was the most predominant year

for NUS classification studies. Nevertheless, research on the spatial distribution of NUS remains sparse and limited to developed regions. The increase in literature could point to an increase in the interest in NUS and the general application of UAV-acquired remotely sensed data.

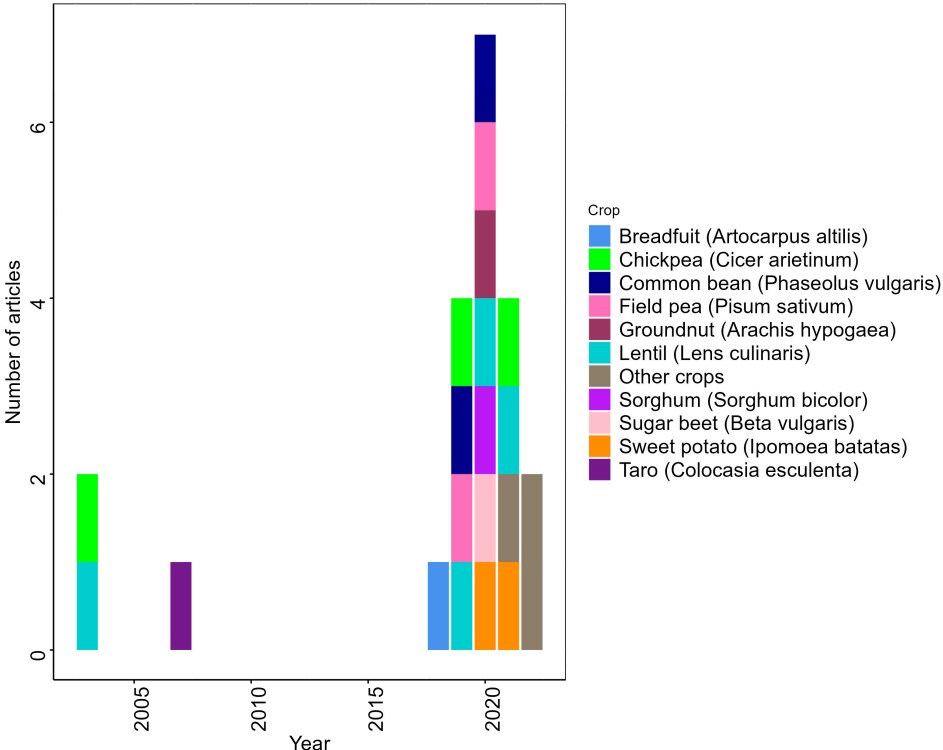

**Figure 8.** Frequency of published articles on NUS classifications.

Fifty articles estimated the stomatal conductance of NUS based on satellite-borne remotely sensed data. Specifically, studies in the USA and China mapped these NUS mostly using sensors, such as Planet scope, analytical spectral devices, UAVs, MODIS, and Sentinel 2 MSI (Supplementary Figures S1 and S2). There was a modest amount of research on stomatal conductance between 2003 and 2011. However, the literature related to farm-scale stomatal conductance increased from 2013 to 2022 (Figure 9). In remote sensing, the stomatal conductance of NUS, sweet potato received more research attention (17 studies), followed by taro (5 studies), cowpea (5 studies), sorghum (4 studies), and amaranth (4 studies) (Figure 9). Meanwhile, very few studies have been conducted concerning the stomatal conductance of crops such as cassava and millet. These crops were less frequent in the retrieved literature (Figure 9a,b) than previously stated. However, there were few studies on remote sensing applications for estimating the stomatal conductance of taro based on UAV-acquired remotely sensed data from the retrieved literature. The specific countries and sensors used in mapping these NUS are detailed in Supplementary Table S6.

*3.3. Types of Sensors and Their Spectral Resolutions*

The utilization of Earth-observation sensors in the remote sensing of NUS studies is significant. Thirteen different sensor types were noted in the reviewed literature (Figure 10). In terms of sensors, the findings of this review revealed that the spectrophotometer was the most widely used sensor for characterizing the health status of NUS, being used in 27 studies. Furthermore, research results indicate that various studies have used hand-held hyperspectral devices to acquire in situ remotely sensed data to detect and map NUS biophysical and phenological attributes. The most predominant sensors in the retrieved literature were spectrophotometers (27), UAV-borne sensors ($n = 18$), spectrometers ($n = 18$), radiometers ($n = 11$), Sentinel- 2 MSI ($n = 9$), MODIS ($n = 3$), and LiDAR ($n = 3$)

(Figure 10a). Meanwhile, the most frequently used drone-borne sensors were RGB cameras (in 11 studies) [26], RedEdge-MX (in three studies) [27], and Canon (in three studies) [23]. These were followed by thermal cameras [28], Parrot Sequoia, Micasense Altum [20], CMOS cameras [29], and MCA6 [17] in order of frequency in the retrieved literature (Figure 10b).

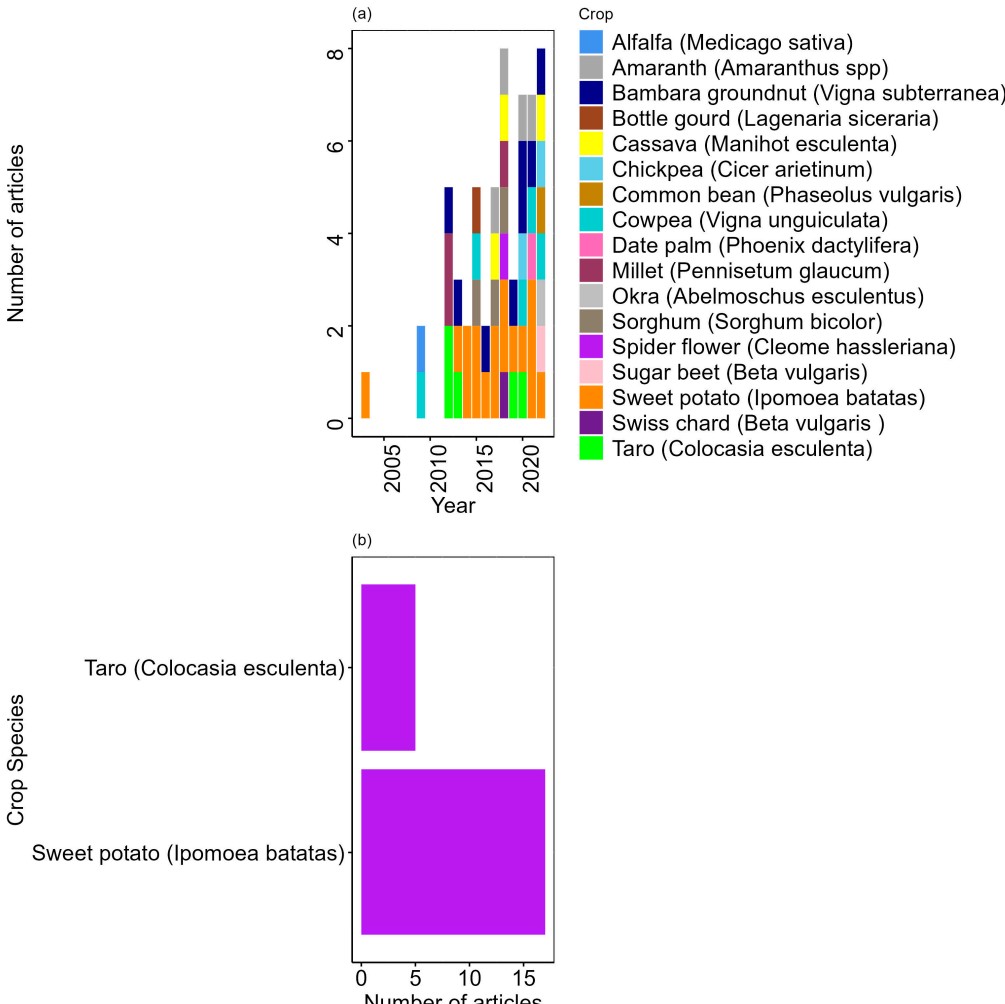

**Figure 9.** Frequency of published articles on mapping NUS stomatal conductance based on (**a**) all sensors for all NUS and (**b**) only on sweet potato and taro.

Across all platforms, the multispectral (broadbands) were highly utilised in the literature compared with hyperspectral (narrow) bands. The visible section of the electromagnetic spectrum, specifically the red, green, and blue (RGB) sections, are primarily the most utilised wavelengths in mapping the spatial distribution of NUS crops and their health attributes (Figure 11). Specifically, the RGB sections of the electromagnetic spectrum (EM) were utilised in 17, 18, and 49 studies based on drone, satellite-borne, and hyperspectral sensors, respectively (Figure 11). The second most widely used section of the electromagnetic spectrum in the literature was the NIR section, utilised in 12 studies with drone-acquired data, 18 studies with satellite-borne sensors, and 38 with spectroscopy. When considering only the drone-borne sensors, seven studies utilised the electromagnetic spectrum's red edge (RE) section. In comparison, 12 studies utilised the satellite remotely sensed RE section, while 44 studies utilised RE bands from spectroscopy. Few studies attempted to engage the thermal bands in characterising the spatial distribution of NUS and their health attributes. Four studies used drone-acquired thermal remotely sensed data and a similar number of studies used satellite-acquired thermal bands. Also, only five studies utilised the spectroscopy thermal section of the EM (Figure 11). When considering

drone-borne sensors, only 2 studies utilised the ultra-violet (UV) and 38 studies used the spectroscopy UV section of the EM.

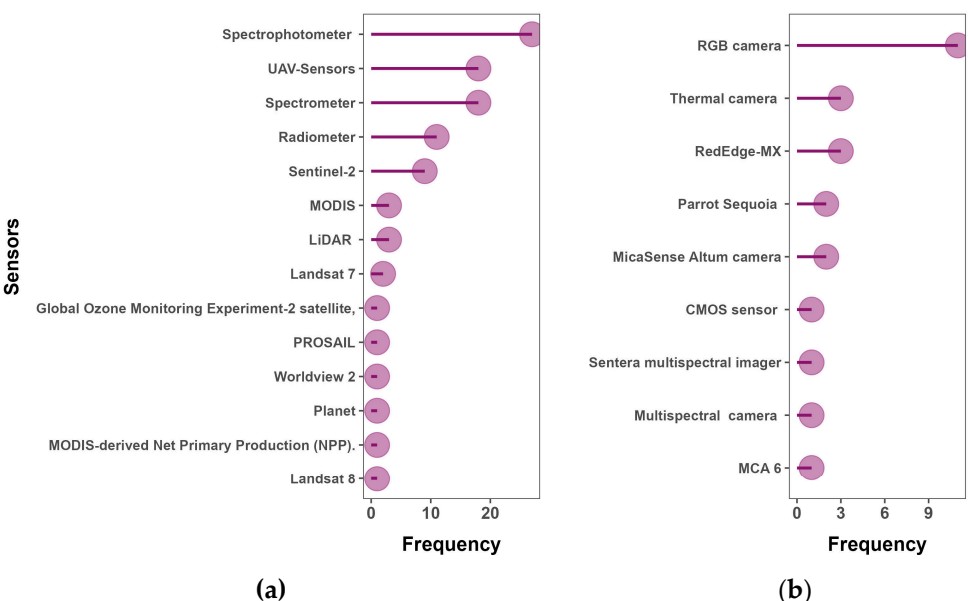

**Figure 10.** Frequency of (**a**) all sensor and (**b**) drone-borne sensors that have been used to map the spatial distribution of NUS and their attributes. (RGB represents red, green, and blue).

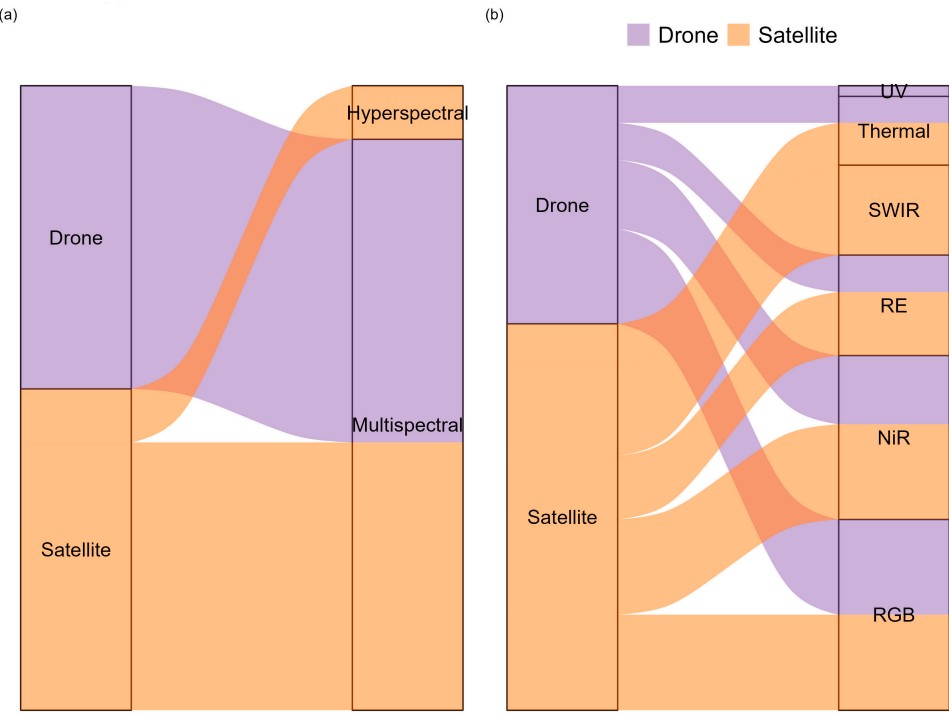

**Figure 11.** Visual distribution (frequency) of (**a**) Spectral characteristics of satellite and drone borne sensor, and (**b**) specific sections of the electromagnetic spectrum they covered in the literature (See Supplementary Table S2 for frequency values). UV is ultraviolet, SWIR is shortwave infrared, RE is red edge, NiR is near-infrared, and RGB is the red, green, and blue spectra. (See Supplementary Materials spreadsheet for frequency values.

### 3.4. UAV Platforms Utilised in the Literature

Regarding the drone platforms, the DJI fleet was utilised in a marginally higher number of studies in relation to all other platforms (*n* = 12). The Octocopter [30], mikrokopter [17],

and Sensfly eBee [6] each featured in separate single studies (Figure 12b). When assessing the frequency of each specific sensor in the retrieved literature, it was observed that the DJI Phantom 4 Pro was the most frequently used platform across the board (appearing in six studies) [23], followed by the Octocopter UAV, which was utilised in one study [31] (Figure 12a). Furthermore, quadcopters were the most widely used drone platform type, followed by fixed-wing drones (Figure 12b). Quadcopter drones were utilised in 16 studies, and fixed-wing drones were used in only 2 studies. Furthermore, the DJI UAVs were popular in mapping a wider range of crops and research domains when compared with other platforms (Supplementary Tables S3 and S4). This could indicate that these platforms are more popular and versatile.

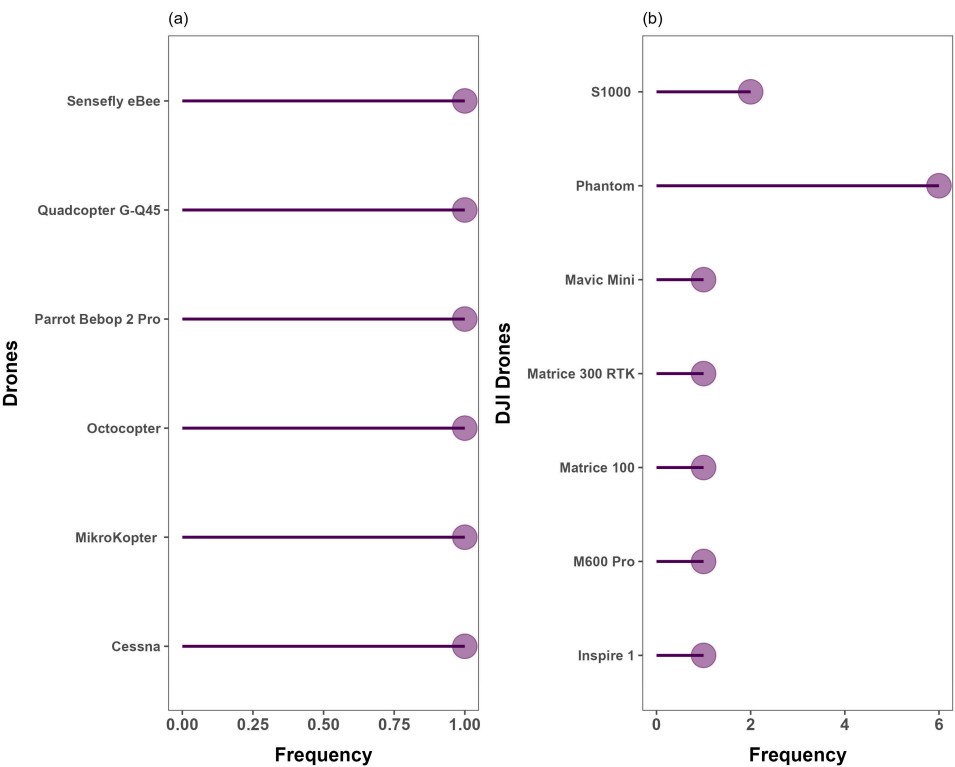

**Figure 12.** Frequency of (**a**) summarised drone platforms (**b**) and DJI drones utilised in the literature to map the spatial distribution of NUS and their health attributes.

*3.5. Derived Vegetation Indices in Remote the Spatial Distribution and Health of NUS Crops*

Algebraic combinations derived from multiple spectral bands, which are commonly known as vegetation indices (VIs), were used to estimate vegetation vigour and vegetative characteristics (canopy biomass, absorbed radiation, and chlorophyll content) in the retrieved literature [32]. Furthermore, the visible (green: 530–570 nm, red: 640–680 nm, and red edge: 730–740 nm), near-infrared (770–810 nm), and red edge (730–740 nm) sections of the electromagnetic spectrum were common in studies that assessed crop health. The reflectance values of these prominent wavelengths are generally used to calculate vegetation indices, such as the normalised difference vegetation index (NDVI), NDVI–red edge (NDRE) simple ratio (SR), green normalised difference vegetation index (GNDVI), green chlorophyll index (CIgreen), and soil-adjusted vegetation index (SAVI), which were frequently used in the retrieved literature (Figure 11). All VIs that were used in the literature are also listed in Supplementary Table S1. In this regard, there is still room to assess more image transformations in mapping the spatial distribution of NUS, such as sweet potato and taro, dominant in smallholder croplands.

### 3.6. Statistical and Machine Algorithms Were Utilised in Mapping the Spatial Distribution and Health of NUS Crops

This study's findings show that several basic statistical procedures, simple regression techniques, and machine learning techniques were used in mapping the spatial distribution and health of NUS crops. These algorithms can be further subdivided into three categories, which are (i) generic GIS classifications, (ii) machine learning and regression techniques, and (iii) multivariate techniques.

Based on Figure 13, the main machine learning algorithms utilised in conjunction with drone-acquired data were linear regression (39%), random forest (28%), support vector machine (SVM) (22%), and artificial neural network (ANN) (17%), in order of frequency in the literature. Linear regression techniques were the most frequent regression algorithms in assessing NUS crop health (Figure 13) [22,30]. The frequency of linear regression and machine learning-based algorithms was also detected during the bibliometric analysis, as illustrated in Figure 2b (red cluster). The RF was the second most widely used machine learning ensemble, followed by SVM, PLS, linear discriminant analysis, and artificial neural network (ANN), in order of frequency in the retrieved literature.

**Machine learning and general regression techniques (MLR)**

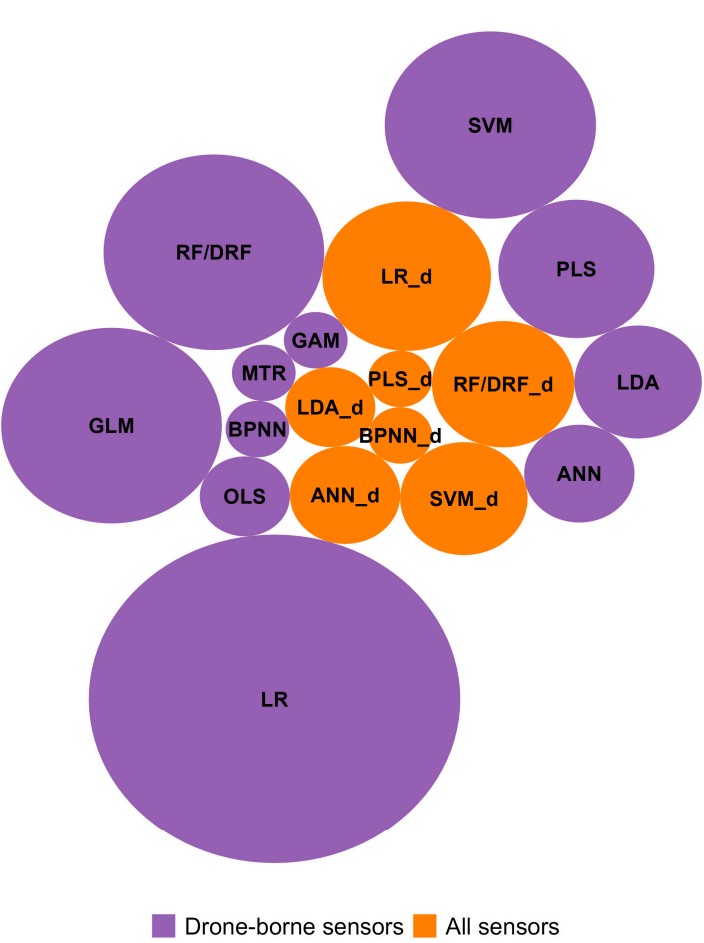

**Figure 13.** Frequency of machine learning and general regression techniques used in remote sensing NUS attributes based on all sensors and drone-borne sensors. (GLM is generalised linear model, RF/DRF is random forest, LR is linear regression, SVM is support vector machine, OLS is ordinary least squares regression, ANN is artificial neural network, BPNN is back propagation neural network, GAM is generative adversarial networks, LDA is linear discriminant analysis, and PLS is partial least squares regression. (See Supplementary Materials spreadsheet for frequency values).

In terms of generic statistics and classification techniques, Pearson correlation, ANOVA, maximum likelihood (ML), OBIA, and the empirical line method (ELM) were the most frequently utilised algorithms based on satellite and drone-borne remotely sensed data (Figure 14). The Mahalanobis distance, parallelepiped, k-means, and canny edge filtering were some of the popular generic classification algorithms used to map NUS based on satellite and drone-acquired remotely sensed data. Studies that were based on generic classification were relatively few for each algorithm (<5) when compared with all other algorithms utilised in the retrieved literature (Figure 14a). Generic classification procedures, such as the analytical hierarchical process (AHP), cluster analysis, fuzzy logic, and multi-layer perceptron, have not yet been utilised in conjunction with drone-acquired remotely sensed data for crop mapping (Figure 14b).

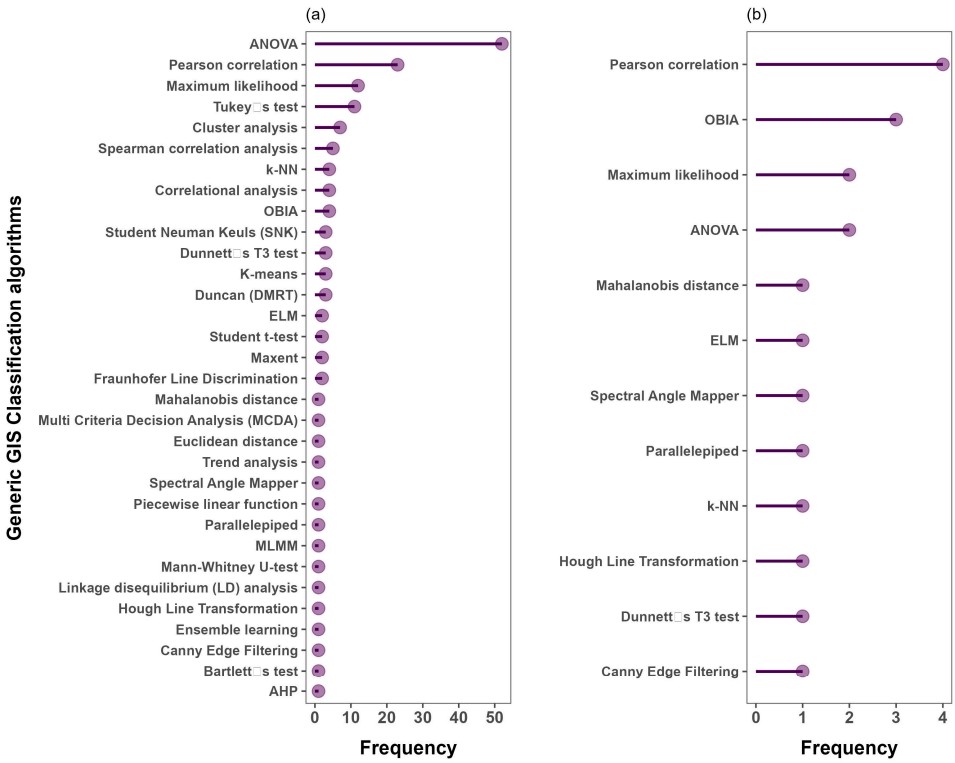

**Figure 14.** Frequency of generic GIS classification techniques used in remote sensing NUS attributes based on (**a**) all sensors and (**b**) drone-borne sensors.

Regarding multivariate techniques, PCA followed by cluster analysis and multiple regression were the most frequently used algorithms based on satellite and drone-acquired data combined (Figure 15). Studies based on linear mixed model and machine learning algorithm classifications were relatively few (<5) compared with PCA and cluster analysis. There seemed to be scanty literature (<5) that utilised multivariate techniques in conjunction with drone-acquired data for mapping the spatial distribution of NUS and their health attributes.

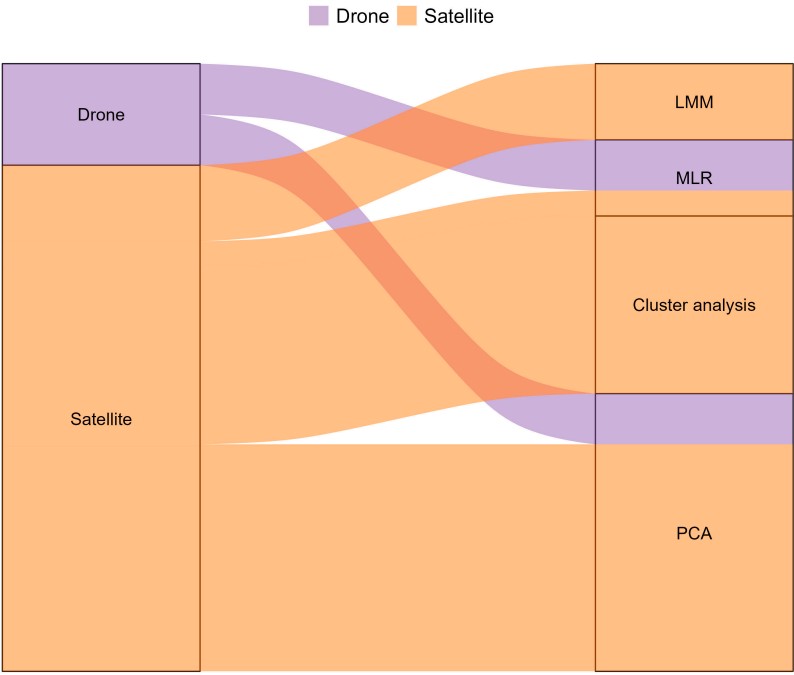

**Figure 15.** Frequency of multivariate techniques used in remote sensing NUS attributes based on all sensors and drone-borne sensors. (See Supplementary Materials spreadsheet for frequency values).

## 4. Discussion

### 4.1. Evolution of Drone Technology Applications in Remote Sensing

There have been numerous shifts in the key terms of the literature on the remote sensing of the distribution and health status of NUS crops. Specifically, there was a shift in the topical terms from mere correlations based on RGB remotely sensed data in mapping NUS attributes between 2018 and 2019 to using hyperspectral data in predicting and mapping NUS crop health attributes, such as yield and AGB, in 2020. Currently, research efforts are being exerted towards the fusion of drone-acquired data with satellite remotely sensed datasets in conjunction with robust machine learning algorithms, such as PLSR and SVM, in characterising yield genotype canopy coverages, amongst others. Progress is discussed in detail in the following sub-sections.

#### 4.1.1. Frequency of Publication and Their Geographic Distribution

This study's findings revealed that the articles that utilised drone-based remotely sensed data in mapping the spatial distribution, health, and productivity elements of NUS crops increased gradually from 2009 to 2014 (Figure 3). This trend was similar to that of published studies that utilised satellite remotely sensed data (Figure 3b). The rapid changes, improvements, and increased accessibility of Earth-observation sensors and platforms could explain this. That period was characterised by limited access to high-spatial-resolution remotely sensed data for crop monitoring. These findings are echoed by numerous reviews, which include Mutanga, Dube [33] and Sibanda, Mutanga [34].

In the context of satellite-borne sensors, the research period from 2009 to 2014 was mainly dominated by the utility of Landsat and MODIS [35]. These sensors were incapable of capturing the land fragmentation and heterogeneity associated with NUS. Moreover, the only accessible fine-spatial-resolution images were those procured from commercial sensors, such as WorldView and QuickBird. These sensors are often associated with exorbitant costs that restrict research activities, especially in under-developed countries. The potential of NUS was still being researched during that period, and research investments were not channelled to the use of satellite-borne sensors. However, the period between 2015 and 2022 was marked by a rapid increase in drone platforms and associated sensors, hence the

rapid increase. This would suggest an increase in the interest in NUS and the improved capabilities of drone and satellite sensors.

Interestingly, most of these drone-related studies were conducted by universities and agricultural institutions in China, America, South America, Europe, and Australia (Supplementary Table S6). This could be attributed to the fact that the earliest drone technologies emerged in these regions between 1849 and 1916 [34]. Also, some of the mentioned countries are pioneers in the research and preservation of NUS. Since then, technology has been spreading and advancing. As a result, there is an increasing need to improve the application of UAV technologies for precision agriculture in accordance with NUS spatial extent and health assessments. Furthermore, it was noted that most of the studies were conducted on experimental plots, in some instances with irrigation facilities at university experimental plots. Generally, commercial farmers who are endowed with resources are the most dormant users of these technologies in under-developed regions, such as southern Africa. No studies were conducted in rainfed smallholder croplands in the retrieved literature on remote sensing NUS. The findings of this study imply that these technologies are slowly being embraced and incorporated from developed countries to the Global South.

4.1.2. NUS Crop Attributes That Have Been Remotely Sensed Using Drone-Acquired Data

The limited number of studies focusing on NUS suggests that the utility of UAVs is still in its infancy in practice. In the retrieved literature, there were no studies that indicated whether there are any spectral libraries that have been generated for NUS. This could imply that, despite the significant efforts in advancing the remote sensing of NUS, more efforts are still required to match with other crops, such as maize. Although the results revealed that, from the year 2020 going onwards, there has been an increase in the number of studies, Merkert and Bushell [36] noted that there are still few research efforts directed towards NUS (Figure 8). In developing countries where NUS are beginning to gain recognition, the limited number of studies could be attributed to the expenses associated with drones and their accessories. Furthermore, the requirements to license and operate a drone in developing countries are still a challenge [37,38]. These findings underscore the importance of increasing knowledge and literature in the context of NUS to improve our insights into them as alternative crops.

The findings showed that 16 studies assessed crop phenology, 9 were on productivity, and 7 explored physiology using drone-acquired remotely sensed data. However, most of the retrieved studies focused on interrelated themes, which explains the high frequencies of studies on each crop attribute. Crop attributes, such as physiology, productivity, and phenology, are mutually dependent elements that determine crop development and health. According to Fageria, Baligar [39], crop physiology in particular is a useful crop attribute that could be used to quantify crop growth for optimising crop yields.

The findings of this study showed that 11 NUS crop productivity elements, namely, crop yield, crop growth, crop health, chlorophyll content, biomass, LAI, canopy height, cover, leaf nitrogen, leaf water content elements, and stomatal conductance, were the most widely researched [28,40]. These are the principal optical crop productivity and health elements; hence, they are anticipated to be covered extensively in the literature on agricultural remote sensing applications. The primary function of remote sensing applications is to optimise yield production and the supply of nutrient-rich foods [41]. Additionally, all research efforts have been exerted towards optimising agricultural productivity (increasing yields) and addressing sustainable goals 1 and 2 on hunger and poverty [23,33,42,43]. Subsequently, yield is expected to be the most intensely researched NUS crop attribute based on drone and satellite-borne remotely sensed data [23,42]. It must be noted that, in most instances, yield is synonymous with AGB. The findings of this study showed that 61% of drone-based remote sensing studies on NUS estimated crop AGB. AGB is an important parameter that can also be utilised to predict crop growth, yield. and productivity [44].

The findings of this study also showed that LAI is one of the most researched elements of NUS, featuring in 33% of the retrieved literature. This can be attributed to the fact that LAI is yet another accurate proxy of plant growth, as illustrated in Figure 7. It has been extensively proven to correlate with other plant growth indicators, including chlorophyll content, biomass, and, to some extent, yield [45,46]. LAI represents the structural attributes of the leaf components estimated by the area of the leaf per unit of ground surface area [47]. The area covered by leaves per unit of the ground surface changes with the changes in a crop's growth stage. This variable can accurately represent the space available for photon interception across a crop's phenological stages, which can affect yield [48]. Subsequently, LAI is a plausible indicator of canopy health and development that needs to be accurately mapped [49]. In addition, LAI can affect the surrounding canopy and the microclimate, e.g., radiation from the sun is intercepted by leaves, affecting transpiration and leaf surface temperature. This ultimately influences the photosynthetic nature of leaves and the stomatal conductance [50]. In this regard, LAI was one of the most important NUS attributes in the retrieved literature.

The chlorophyll content is another predominant NUS attribute covered by 50% of the retrieved literature in this study [51,52]. The chlorophyll content of leaf tissue indicates a plant's physiological structure, nutritional composition, and health [53,54]. The antenna pigments in chloroplasts absorb incoming solar radiation during photosynthetic activities [55]. The resulting radiation is then transferred to the reaction centre pigments, which discharge electrons to activate the photochemical process [55]. Specifically, chlorophyll is a very relevant and optical indicator of crop health, since chlorophyll highly absorbs energy in the red (650–700 nm) and blue (400–500 nm) regions of the electromagnetic spectrum to increase photosynthesis. The types of chlorophyll responsible for the higher absorption within the visible spectrum are chlorophylls a and b [55]. In this regard, there is a positive relationship between the leaf total chlorophyll content (chlorophyll a+b), the solar radiation absorbed by the leaf tissues, and the photosynthetic rate of the crop [55]. Subsequently, a leaf's biophysical pigments and biochemical photosynthetic processes are linked to a plant's health and productivity. Hence it is considered a suitable proxy for crop health in the light of agricultural remote sensing applications [54].

Meanwhile, the chlorophyll content has been widely proven to correlate positively with the nitrogen content in various crop species [56–59]. Indeed, approximately 75% of the total nitrogen is stored within the leaf chloroplasts [60]. In this regard, the findings of this study revealed that 28% of the retrieved studies estimated nitrogen concentration using drone remotely sensed data, thus rendering the nitrogen content another sought-after attribute of NUS. An elevated nitrogen concentration is associated with an increased CO2 assimilation rate and stomatal conductance in the crop, which aids in producing chlorophyll and green pigments. For example, Muhammad, Alam [61] conducted a related study to evaluate the impact of various nitrogen and phosphorous levels and beneficial microbes on enhancing canola productivity. The results revealed that nitrogen applied at a rate of 180 kg ha$^{-1}$ increased plant pods, seed pods, the seed-filling duration, seed weight, biological yield, and seed yield.

The findings of this review also showed that crop structural parameters, such as crop height, crop growth, and canopy cover, were explored in 39%, 72%, and 39% of drone-based studies, respectively. These crop structural parameters are related to AGB, LAI, and the chlorophyll content, which are measured and used to predict the yield. Hence, these parameters are directly linked to food and nutrition security. The crop height and growth attributes directly interact with the incident electromagnetic energy that is typically measured and used to model productivity. Furthermore, these physiological crop variables are sensitive to variations in environmental conditions. These include environmental (precipitation, temperature, soil type, etc.) and biochemical conditions (fertility, weeds, pests, and diseases). In this regard, these structural attributes are instrumental in monitoring the health of crops to optimise crop production; hence, significant research efforts were devoted to them.

This study showed that stomatal conductance was another optimal NUS health parameter assessed by 11% of the retrieved studies that utilised drone data. The stomatal conductance measures the degree of stomatal opening and can indicate leaf gas exchange [62]. The stomatal conductance is strongly associated with leaf transpiration rates, photosynthetic efficiency, chlorophyll concentration, and nitrogen concentration [63]. Specifically, periods of crop water stress or drought stress will be limited by the $CO_2$ concentration at carboxylation sites (Cc) inside the chloroplast. This is determined by the $CO_2$ diffusion components, i.e., stomatal conductance (gs) and mesophyll conductance (gm) [64]. Particularly, the stomata control the $CO_2$ diffusion into the leaf tissue and water diffusion out of the plant. Therefore, it has been proven that, under water deficit conditions, plant stomata will close to prevent major water loss. This consequently decreases photosynthesis via the decreased influx of $CO_2$ [64].

In this regard, higher stomatal conductance and high photosynthetic efficiency are associated with limited moisture stress. Hence, these plants will be healthier, with a higher chlorophyll content and green pigment. In this regard, stomatal conductance was among the most researched crop attributes in the retrieved literature because it is an accurate proxy of moisture stress and the health status of crops [28,62]. In fact, the stomatal conductance aids in water use and irrigation scheduling as a pathway towards optimising food production. A study by Chai, Massawe [65] measured the morpho-physiological traits of Bambara groundnut exposed to progressive mild drought in a controlled environment. Drought stress reduced stomatal conductance significantly ($p < 0.01$). Furthermore, higher stomatal density and reduced leaf area were observed in drought-treated plants ($p < 0.01$). This suggests that NUS crops are more resistant to biotic and abiotic stresses, such as drought and water stress. Above all, this indicates the prospects of using stomatal conductance as a proxy for understanding crop water stress.

In most studies from the retrieved literature, there were strong correlations between the stomatal conductance and water relation status or leaf water content [28,66,67]. Specifically, 17% of studies based on drone remote sensing researched the NUS leaf water content. Water content is an important indicator of crop health [28,66,67]. A plant with higher water potential will produce greener pigmentation and have increased crop productivity. According to Ouyang, Struik [64], under water deficit or conditions of mild water stress, the stomatal conductance, internal $CO_2$ concentration, and net assimilation rate within a plant will decrease, and the A/G ratio will increase. Various studies assessed and reported that crop water stress allows for a decline in the total chlorophyll of various crops and a decline in plant productivity [68–70]. This includes non-stomatal regulation of photosynthesis, a decline in light and the $CO_2$ concentration, a reduction in photochemistry, declining activity of photosynthetic enzymes, and lowered mesophyll conductance. For instance, Chibarabada [71] assessed the stomatal conductance of three grain legume crops (groundnut, dry bean, and Bambara groundnut) grown under three water treatments. Their results indicated that, under varying water regimes, NUS crops adjusted to constrained soil water through stomatal regulation and reduced canopy size. Furthermore, Bambara groundnut showed a positive attribute under water-limited conditions. It had the lowest stomatal conductance under all watering regimes compared with the other crops. In the context of remote sensing, these plant conditions then impact the interaction between different sections of the EM spectrum and the various magnitudes of crop water stress [72]. This example illustrates why stomatal conductance and foliar temperature were significantly considered in NUS production.

### 4.1.3. Sensors and Platforms That Were Used in Remote Sensing NUS

The dominance of drone-based remote sensing studies on NUS could be explained by the fact that most of these crop species are orphaned, neglected, and underutilised. In this regard, they are generally planted in smaller areas than mainstream crops, such as maize. On the other hand, very few freely accessible satellite-borne sensors offer finer-spatial-resolution data suitable for capturing the variety of crops in fragmented smallholder

croplands. Subsequently, the freely available datasets from moderate-spatial-resolution sensors, such as Landsat and Sentinel 2 MSI, cannot capture the dynamics of crops in smallholder farms when compared with UAVs. Drones offer rapid, ultra-fine-spatial-resolution data, often to a sub-metre resolution, in a cost-effective manner [73].

Furthermore, the user determines the spatial and spectral resolution of the drone remotely sensed data, offering endless opportunities in crop phenotyping from the stand level to field scale. Sentinel-2 MSI was the most widely used satellite-borne sensor in mapping the spatial distribution, health, and productivity parameters of NUS crops. Sentinel 2 MSI has a minimum spatial resolution of 10 m and a spectral resolution that covers the red edge section of the electromagnetic spectrum, which is sensitive to crop health [74]. These attributes make Sentinel 2 MSI the second most suitable sensor among UAV-borne sensors in mapping NUS, often grown in small, fragmented fields.

In terms of the drone sensors, our findings showed that the RGB and multispectral cameras (>three bands) were the most frequently used in the retrieved literature when compared with hyperspectral sensors (Figure 11).This could be explained by the exorbitant expenses associated with hyperspectral sensors to multispectral cameras. RGB and multispectral cameras generally cover the visible, including the red edge and the NIR, sections of the EM spectrum. This is usually in not more than six broader spectral band resolutions (i.e., MicaSense Altum) [75]. Meanwhile, most hyperspectral sensors, such as the Cubert S185 hyperspectral sensors, cover a wavelength range between 450 and 950 nm. These are composed of 125 channels at a resolution of 8 nm at 532 nm, ranging from visible to near-infrared (450–950 nm) with a sampling interval of 4 nm [76]. Furthermore, the RGB type of sensors offers models with less accuracy when compared with multispectral or hyperspectral images covering a wider spectral range [77]. Hyperspectral sensors have been widely proven to offer robust narrow, relatively contiguous bands, although multicollinearity issues often impact them. Despite their exorbitant prices, these bands are more sensitive to subtle crop variations compared with broadband sensors [78–80].

The findings of this study also showed that very limited studies utilised cameras that captured data in the red edge section of the EM spectrum. The multispectral sensors that captured the red edge section in the retrieved studies were the MicaSense series, and Red-edge MX utilised in two and three studies, respectively. The red edge section has been vastly proven to be sensitive to minute variations in plant attributes associated with health and productivity, such as the LAI, chlorophyll content, stomatal conductance, AGB, and nitrogen content [49]. For instance, the increased chlorophyll content, LAI, and biomass generally result in increased absorption in the red region, pushing the red edge to longer wavelengths [81]. Considering this phenomenon, the red edge has become one of the most sought-after sections of the EM spectrum in crop monitoring [82]. In this regard, there is a need for more studies that assess the robustness of hyperspectral and red-edge sensors in mapping NUS attributes. This will improve the assessment and monitoring of the health elements of NUS if their productivity is optimised based on the information derived from remotely sensed data.

The findings of this study also showed that the multirotor drones were the most widely used platforms in the retrieved literature compared with fixed-wing drones (Figure 12). This could be explained by the fact that, despite the endurance associated with fixed-wing drones, they often require take-off and landing space. However, these requirements are not always available in experimental sites, where most retrieved studies have been conducted [83]. Multirotor drones are characterised by vertical take-off and landing (VTOL), which makes them more suitable for utilisation in research areas with limited take-off space [83]. These drones are relatively cheaper than fixed-wing drones [34]. This could also explain the high utilisation frequency of DJI drones in the retrieved literature (Figure 12). The advantage of DJI drones is that some are associated with an automated image acquisition procedure, making it easy to fly them, as they require less expertise in drone piloting. According to Sibanda, Mutanga [34], the DJI platforms are generally compatible with many

types of sensors from other platforms, which could also explain their wide utilisation in studies on the remote sensing of crops.

### 4.1.4. Performance of Vegetation Indices, Classification, and Estimation Algorithms

This review's findings also showed that vegetation indices (VIs), such as NDVI, NDRE, SR, GNDVI, CIgreen, and SAVI frequently mapped the spatial distribution of NUS and their health parameters. For instance, Li, Zheng [84] deployed a quadcopter UAV to investigate its utility in crop identification from different land cover types based on VIs. The maximum likelihood classifier, in combination with optimal VIs, exhibited high classification accuracies in that study. This approach was accurate because VIs could partially overcome the influence of shadows and other noise in the background [85]. A growing body of literature demonstrates the optimal performance of VIs [22,24,29]. Specifically, VIs are robust because they can suppress background effects compared with bands [79]. Furthermore, VIs derive their strength from two or more sections of the EM spectrum. For instance, Vis, such as NDVI, NDRE, SR, GNDVI, CIgreen, and CIred-edge, derive their strength from sections of the EMS (i.e., red edge, NIR, and SWIR) that are more sensitive to crop elements, such as the chlorophyll content, LAI, ABG, nitrogen concentration, and stomatal conductance [53]. Moreover, a large body of literature has proven that red-edge-based VIs are more robust than conventional vegetation indices, such as NDVI and SR [20,86]. NDVI tends to be insensitive to increases in chlorophyll, LAI, and biomass [85,87]. As aforementioned, the red edge is highly associated with plant physiological traits, such as the leaf angle distribution, chlorophyll concentration, and LAI, which directly influence vegetation spectral reflectance. Subsequently, red edge-derived vegetation harnesses this robust sensitivity to variations in LAI biomass and chlorophyll content, among others [85,88–91].

As was suggested in many studies, the combinations of sensitive spectral variables with robust algorithms produce models with relatively optimal accuracies [92]. The findings of this study showed that the machine learning algorithms that were utilised in conjunction with drone-acquired remotely sensed data were primarily linear regressions (LR) (39%), followed by RF (28%), SVM (22%), and ANN (17%). Machine learning regression algorithms present a potential approach for generating adaptive, robust, and fast crop estimates. A growing body of literature demonstrates machine learning algorithms' efficiency and optimal performance in estimating crop biophysical parameters [47,93,94]. The high frequency of using regression models, such as RF, LR, and SVM, can be credited to the fact that they are simple to implement using various platforms. This ranges from Microsoft Excel to complex programming platforms, such as R Statistical Software and the Google Earth Engine platform. Despite their high frequency in the retrieved literature of this current study, LR models are parametric algorithms associated with data normality assumptions. However, these assumptions are often challenging to attain due to high spatial varieties in crop fields. Furthermore, these models are susceptible to overfitting and outliers [95]. Subsequently, more efforts have been channelled towards machine learning algorithms, such as SVM and RF. These algorithms are non-parametric, robust, and less prone to outliers, as they are not extremely affected by the data frequency distributions [95]. Specifically, RF has (i) hyperparameters that are easy to adjust, (ii) is robust to outliers, overfitting, and data dimensionality, (iii) low bias and moderate to minimal variance due to the averaged trees, (vi) works well both continuous and categorical variables, (v) has a capability of discerning the importance of predictor variables, and (iv) it is relatively resistant to multicollinearity when tree depths are greater [91,96,97]. This makes this algorithm suitable for mapping cropping attributes.

Other than machine learning algorithms, numerous generic classification algorithms were utilised in the retrieved literature, such as maximum likelihood and minimum distance to mean classification algorithms [98–100]. Despite their optimal performance in the literature, these algorithms are less robust when compared with machine learning algorithms. For instance, maximum likelihood is also a traditional parametric algorithm

that is often difficult to parametrise, despite the fact it is generally accessible through freely available GIS platforms.

### 4.2. Challenges in Mapping the Spatial Distribution and Health of NUS Using UAVs

Even though there is progress regarding the utility of drones in crop mapping and health quantification of NUS, several challenges impede their propagation, especially in Africa. Many African countries are battling to address various drone-related issues, such as privacy, public safety, and preventing the possession of malicious drones [101]. Moreover, restrictive UAV or drone regulations across many developing regions, including Africa, hinder their utilisation. To utilise UAVs, users must seek permission from landowners and municipalities, and only operate in certain areas, among several other issues [102].

For example, the civil aviation authorities (CAAs) in many countries aim to prevent UAVs from entering the flight paths of manned aircraft. Moreover, CAAs are attempting to construct an inclusive system that accommodates UAVs into their respective air navigation and surveillance systems [103]. To the best of our knowledge, only a few African countries, including Ghana, Kenya, South Africa, Rwanda, Zimbabwe, and Tanzania, have established complex legal requirements governing the use of UAVs in varied airspace practices. Some drone restrictions in South Africa (SA), for example, state that UAVs weighing more than 7 kg (15.4 pounds) are not permitted to operate [102]. In this regard, the regulation on the mass of UAVs at taking off tends to indirectly restrict the areal extent and the size of the camera to be mounted for research purposes [34]. Specifically, due to the weight restrictions, many of the sensor types that are frequently used tend to be lightweight, small, and general consumer grades with limited spectral resolution [34]. Also. this could explain why more studies are based on the VIS (RGB) spectra than other sections of the electromagnetic spectrum.

Furthermore, SACAA asserts that drone operators should maintain a continuous visual line of sight (VLOS) with their drones during flights. Moreover, remote-piloted aircraft (RPAs) are not permitted to fly beyond visual-line-of-sight in designated places (BV-LOS) [102]. Sibanda, Mutanga [34] stated that supporting regulations and the operationalisation of BVLOS drone technology applications will facilitate coverage of greater areas on a single mission. Covering a greater area in a single mission improves the cost-effectiveness of acquiring VHR imagery. Other restrictions include requiring drone operators to obtain an RPAS operator certificate (ROC) from the CAA before flying [104]. UAV operators should obtain insurance to cover their liability in the case of committing physical or bodily harm to another individual while operating their drone [104].

Aside from these regions, numerous African countries are still attempting to establish the necessary regulations that endorse UAV operations. As a result, these regulations are becoming increasingly complicated, while attaining RPAS is associated with high expenses. More notably, the cost of obtaining a drone pilot license is excessive, with estimates ranging from USD 1500 to USD 2000 in 2021 [34]. Furthermore, the value of drone platforms, including sensors, is prohibitively expensive for several minority groups or scholars, making these technologies unattainable for research purposes in the vast bulk of Sub-Saharan African countries [34].

### 4.3. Research Gaps and Opportunities

The following gaps were noted in assessing the utility of drone-based remotely sensed data in mapping the spatial distribution and health of NUS crops:

- The observation of NUS crop health has garnered minimal research attention and interest from the scientific community. Further, few studies have sought to evaluate the utility of drone technology for characterizing crop dynamics, especially in the Global South. The limited research within this region means there are opportunities to innovate;
- Although NUS crops reportedly resist abiotic stresses, such as drought and heat stress, most of this information is anecdotal and inconsistent [1]. This incomplete body

of knowledge around drought and heat stress makes applying and validating RS techniques challenging. Hence, there is a requirement to generate more empirical information on the ecophysiology and morphology of NUS;

- Only a few research studies have sought to evaluate the effectiveness of robust ML algorithms in conjunction with VIs in predicting the spatial distribution and health of NUS crops. Further to this, few studies have attempted to assess and leverage the potential synergies between drone and satellite-borne datasets, especially considering the release of the freely accessible Planet Scope Sentinel 2 MSI and Landsat series;
- The application of UAV-based technology for estimating NUS' spatial extent and health has not attracted significant attention from the geospatial research community in practice. The spatial extent of NUS crops can be predicted at a granular scale using UAV-based modelling and classification. Such models will be useful for predicting crop yield, crop monitoring, predicting soil quality, and modelling evapotranspiration, precipitation, drought, and pest outbreaks;
- Modelling and predicting vegetation key variables, such as LAI, stomatal conductance, and AGB, are critical to understanding and quantifying NUS' morphological and phenological processes in the face of climate change;
- Optimal VIs, such as NDVI NDRE, and VARI, can aid smallholder farmers in analysing trends in plant health. Moreover, NDRE is useful in determining vegetative vigour late in the growing season.

It is, therefore, essential to research the prospects of these remote sensing technologies in monitoring these crops to promote their inclusion as mainstream crops within the agriculture sector.

### 4.4. Way Forward: Closing the Gaps in the Utilisation of Drone Technology in Mapping Spatial Distribution and Health Status of NUS Crops

Scholarly attention needs to be paid to promoting the application of UAVs for assessing NUS crops' spatial extent and health at the field scale. There is a need to increase and extend research efforts towards rainfed smallholder croplands in remotely sensing NUS. Hence, this will optimise their productivity while strengthening the livelihood and food security of marginalised subsistence farmers.

UAVs are becoming increasingly common in agricultural research as relevant sources of high-temporal-resolution data for agricultural health monitoring, amongst other applications. This has been previously limited to a farm scale. UAVs provide immense opportunities by incorporating climate-smart and precision agriculture into smallholder farming. They deliver high-resolution imagery at user-defined temporal resolutions, which benefit crop health monitoring and independent decision-making. For example, the integration of multi-rotor drones, such as the DJI series, in conjunction with multispectral sensors, such as the Mica Sense RedEdgeMX Parrot Sequoia and MicaSense Altum cameras, has been demonstrated to deliver accurate models of crop health characteristics at the field scale.

Additionally, the increasing adoption of cutting-edge narrow-band hyperspectral and LiDAR remote sensing technologies could provide the opportunity to assess diverse crop health parameters at high optical resolutions. This assessment would also allow for creating various optimal VIs. Hence, these are directly associated with NUS crop morphological and physiological characteristics, allowing for the quantification of their health status.

Moreover, to improve research based on the application of UAVs to estimate NUS crop health, it is essential to identify and implement robust and reliable non-parametric ML algorithms. Algorithms that enhance prediction accuracies with fewer assumptions, such as the RF, seem to hold endless prospects. Using multi-fusion techniques based on the combination of ideal VIs and ML algorithms will achieve the best optimal accuracies. As a result, additional research is required to evaluate the utility of UAVs with distinct spectral characteristics for NUS. It would be valuable to research whether UAV sensors that measure spectral reflectance along the electromagnetic spectrum's thermal, SWIR,

and NIR sections enhance the prediction of NUS water stress at the farm scale. Therefore, near-real-time-temporal-resolution spatially explicit information on NUS spatial extent and health can certainly assist smallholder farmers. This will assist in detecting variations in crop morphology to optimise yields. Near-real-time fine-resolution NUS information could be used to draw up early warning systems for smallholder farmers to prevent any damage or reduction to crop yields.

## 5. Limitations of This Study

Many possible sources of information on the spatial extent and health of NUS were disregarded throughout our search approach. We primarily searched for English-language sources and peer-reviewed articles included in Scopus, Web of Science, and Google Scholar. In conducting the literature review, some studies were inaccessible in full length and others were not written in English. This could have had a potentially negative effect on quantifying all the studies that focus on remote sensing the spatial distribution and health status of NUS crops. More so, the exclusion of these studies has an impact on the spatial distribution of NUS studies. Since NUS crops research is highly under-represented and anecdotal, it was challenging to obtain research that focused on the health and spatial extent of NUS crops. Despite missed contributions, we believe that our findings are based on a sample that is adequate and diverse enough to offer meaningful insight and implications in the application of UAVs in the remote sensing of NUS crops in smallholder croplands. Another, limitation of the study was the fact that we limited our scope of research materials for the systematic review to only the Scopus, Web of Science, and Google Scholar databases. As such, we recommend that future work extend the sources to a larger number of databases.

## 6. Conclusions

The objective of this study was to systematically assess the literature on the progress, challenges, gaps, and opportunities in using drone-derived remotely sensed data to map the spatial distribution and health status of NUS. This study placed a special focus on classifying and measuring stomatal conductance in smallholder croplands in the Global South. From the reviewed literature, UAVs emerged as cutting-edge technology with a high potential to provide spatially explicit, timely, and reliable data for assessing the health of NUS. However, there is a noticeable scarcity of studies that have attempted to map the spatial distribution and stomatal conductance, along with other health attributes, of NUS in comparison with mainstream crops in smallholder croplands in the Global South. Several factors contribute to this scarcity, including the limited popularity of NUS, the prohibitive cost of drones and pilot licenses, a shortage of personnel with the necessary skills, and stringent regulations governing drone procurement and operation, among others. While acknowledging the substantial ownership costs associated with drones, our study advocates for the communal ownership of UAVs among smallholder farmers, which can significantly reduce operational expenses. This approach offers promising opportunities to integrate climate-smart agriculture into local farming systems through enhanced monitoring and mapping of crop health. It will equip farmers and, through extension, workers with valuable information for making informed decisions in the field. Additionally, it will enhance stress detection, improve irrigation scheduling, and boost crop productivity on a farm scale. Furthermore, improving the productivity of NUS and other crops presents an opportunity to revive local food culture and food systems, while simultaneously empowering smallholder farmers, who are often the stewards of this agrobiodiverse resource system. We also advocate for the use of UAVs to engage younger generations in agriculture, a critical priority in Sub-Saharan Africa. By delivering high-quality imagery and automating data collection, processing, and analysis at a low cost, the application of UAVs in local food systems can serve as a pathway to optimise food production among disadvantaged smallholder farmers.

**Supplementary Materials:** The following supporting information can be downloaded at: https://www.mdpi.com/article/10.3390/rs15194672/s1. A list of the articles used in this study is provided as supplementary data, Frequences. References [6,12,14,15,17,18,20–24,26–28,30,31,52,66,67, 84,86,94,99,105–123] are cited in the supplementary materials

**Author Contributions:** Conceptualization, M.A. and M.S.; methodology, M.A.; software, M.A. and M.S.; validation, M.A., M.S., T.D. and T.M.; formal analysis, M.A.; investigation, M.A. and M.S.; resources, M.S.; data curation, M.A.; writing—original draft preparation, M.A.; writing—review and editing, T.D., M.S., V.G.P.C. and T.M.; visualization, M.A. and M.S.; supervision, M.S., T.D. and T.M.; project administration, M.S.; funding acquisition, M.S. All authors have read and agreed to the published version of the manuscript.

**Funding:** This work was funded by the Water Research Commission of South Africa (WRC) through Project C2022/2023-00930 titled "Unmanned aerial vehicle high-throughput phenotyping of neglected and underutilized crops species (NUS) for improved water use and productivity in smallholder farms".

**Data Availability Statement:** The data will be availed upon request from the first author. The first author's contact details are detailed in the names section of tis manuscript.

**Conflicts of Interest:** The authors declare no conflict of interest.

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
