# Peer review of "A Systematic Review of UAV Applications for Mapping Neglected and Underutilised Crop Species’ Spatial Distribution and Health"

_remotesensing, doi:10.3390/rs15194672_

Round 1
Reviewer 1 Report (Previous Reviewer 1)
Based on three databases, Google Scholar, Scopus, and Web of Science, this study reviewed the progress, opportunities, challenges, and associated research gaps of UAVs equipped with multi-spectral sensors in the spatial distribution and health assessment of NUS. This paper comprehensively considers the research area, the NUS variety, the biological significance of the study, sensors, and the research algorithm adopted, etc. The limitations of the paper are also discussed. The discussion perspective is comprehensive, and to a certain extent, it provides research ideas and directions for the subsequent research on NUS varieties. However, the following points still need to be modified:
1. Check the format of the references
2. Line 167: check the layout of Figure 1.
3. Lines 274 ~277, Figure 2. What each line color means can be illustrated in the note. In the original text, a large paragraph describes the relationship between these words, but if a simple illustration in the note is added, it will be more convenient for readers to understand.
4. Many data sources in the text should be identified with references, such as line 302-307.
5. Line 325, Figure 5. I can only see the two figures (a) and (b), and the figures (c) is not visible, but it contains the figures (c) in the description.
6. Line 331~333 &357, Figure 6. “Five studies discriminated crops based on drone-acquired data, while six articles were based on satellite-borne remotely sensed data” is hard to understand, because two different colors each have six lines. How should Figure 6 be interpreted? In addition, figures (a) and (b) are used in the comments to Figure 6, but are not marked.
7. Line 385, Figure 8. The colors used by the Breadfuit and Chickpea crops in the figure are too similar to distinguish them. The same problem exists with color matching in Figure 9.
8. Line 414~415 & 426, “…the findings of this review revealed that the hyperspectral spetrophotometer emerged as…”, only “spectrophotometer” is used in the coordinates of Figure 10 (a), it is recommended to use the full name or its abbreviation to highlight the hyperspectral; In addition, the two figures in Figure 10 are marked as (a), and the one on the right needs to be changed to (b).
9. Line 460: ”The Octocopter in 1 studies”:Suggest checking and modifying here.
10. Line 582-585: There is no causal relationship in this paragraph
11. Line 637: “LAI is yet another accurate proxy of plant growth”. Why is another?
12. Line 756~759, What is the number of this figure? No illustration.
13. Line 902: What’s crop phenomena?
14. Check the text near fig 16 and the picture title
15. Discussion: Since all sensors and dron-borne sensors have been mentioned many times in the previous chapter, it is suggested to add satellite sensors to the discussion.
The whole text needs to be carefully checked for spelling and language organization
Author Response
Thank you for your detailed comments reviewer. We appreciate the reviewer's feedback. We believe these suggestions have significantly improved our manuscript.
We have incorporated these suggestions to improve the manuscript's readability, grammar, and layout. The adjustments made have been highlighted in red.
Please see the attachment below.

Reviewer 2 Report (Previous Reviewer 2)
The resubmitted text includes modifications. The changes contemplated all the questions presented in the previous revision.
Author Response
Thank you for your response reviewer. We highly appreciate your positive feedback and ideas. These have greatly improved our manuscript.
Reviewer 3 Report (Previous Reviewer 3)
A Systematic Review of Literature on the Utility of UAVs in Mapping the spatial distribution and health status of Neglected and Underutilised Crop Species (NUS)
Dear Authors
The basic science of this paper is conducted in a good way and is of an appropriate standard. The author and his team write this paper according to the journal's scope and modern trends. I am glad to review this paper because the paper is very interesting according to my research interest area. This study systematically reviewed the literature on remote sensing (RS) the spatial distribution and health of NUS, evaluating the progress, opportunities, challenges, and associated research gaps. 171 peer-reviewed articles were accessed from Google Scholar, Scopus, and Web of Science databases and analysed following the PRISMA approach. The findings of this study revealed that the United States (n =18) and China (n =17) have been the primary locations where studies on remote sensing the NUS crops attributes were conducted, with a few additional studies emerging from regions in the global South, including southern Africa. The author should revise the whole manuscript and fix some grammar mistakes and split long sentences. Moreover, the paper is well-structured. I am going to recommend minor revisions at this stage. I hope, the author will follow our comments and enhance their study and resubmit again in this journal.
Best Regards
Title
•I am not satisfied with the title of this study. Title is very lengthy
Abstract
•The author should revise the whole abstract and explain the whole study in a good way.
Introduction
The introduction part is very lengthy. The author should remove some irrelevant material from the introduction section. Objectives are not clear at the end of the introduction section
2. Materials and Methods
The author should read some review papers and recheck the structure of the paper.
Modify Figure 1.
Its better the author mention proper heading instead of phases. Phase 1=2.1. Literature Search
Phase 2:
2.2.: Data Extraction
Phase 3
2.3: Data analysis
3. Results
Results are appropriate but there are many figures, its better to add some figures in the supplementary material
5. Discussion
I agreed
6. Conclusion
The conclusion part is very lengthy, try to rewrite and highlight the significance of the research.
Author Response
Thank you for your suggestions. We highly appreciate your feedback and ideas. These have greatly improved our manuscript. The adjustments made have been highlighted in green.
Please see the attachment below for your perusal.

Round 2
Reviewer 1 Report (Previous Reviewer 1)
1. There are duplicate named tables in the supplementary data, it is recommended to check.
2. Please check the annotations and formatting of the supplementary data.
I have carefully reviewed the revised manuscript titled "A Systematic Review of UAV applications for Mapping Neglected and Underutilised Crop Species’ (NUS) spatial distribution and health" in response to the first review. I would like to commend the authors for their diligent efforts in addressing the concerns raised during the initial review. The revisions made have significantly improved the quality and clarity of the manuscript.
Author Response
Thank you for your suggestions. We highly appreciate your feedback. These have greatly improved the format of our supplementary data section. The adjustments made have been highlighted in red

This manuscript is a resubmission of an earlier submission. The following is a list of the peer review reports and author responses from that submission.
Round 1
Reviewer 1 Report
This study systematically reviews the literature on remote sensing the spatial distribution and health of NUS, evaluating the progress, opportunities, challenges, and research gaps associated with them. The research direction of the article is very meaningful, but there are problems in the research methods and ideas of the article, the sentence expression is chaotic, and many words in the sentence are inappropriate. I have made a list of my concerns:
1. Line 24:“The rapid advancement in remote sensing (RS) technologies such as unmanned aerial vehicles (UAVs) equipped with ... for plot-level crop analysis required in optimising production”, please check for the syntax error.
2. Line 29:“Subsequently”, there seems to be a different logic problem here.
3. Line 33: “The most remotely sensed NUS crop attributes are crop yield”, please check and correct the sentence grammar error.
4. Line52: “Hence, the decline in water resources and land has increased the pressure on this sector, especially within smaller-holder farms in ensuring long-term food supply for growing populations.” The semantic connection of this sentence is inappropriate.
5. Line 58: “Hence, emphasising the magnitude of food insecurity dilemmas in SSA.” Please check for a syntax error.
6. Line 59: It's best to mark what “SSA” refers to in the front in advance.
7. Line 60-68: Only SSA is mentioned in the introduction, which is not systematic。
8. Results of part of the content is too chaotic, affecting the reading.
9. Line 218:Content missing.
10. Line 224: As Figure 1 has already appeared before, please check and modify the picture number.
11. Please change the size and format of pictures to fit the page and read.
12. Pictures do not have the same title format, please keep the format of each title consistent, preferably to make them center.
13. The reference quoted in the paragraph shall be marked with the specific picture number.
14. Line 309: “ 4.1 ” Please check the number of the title, with the same problem below.
15. Line555-574: The paragraph and the following are inconsistent with the previous format
16. Line575:Content missing.
17. Lines 1011-1012, the article states that these NUS crops' responses to non-biological stresses such as drought and heat stress are anecdotal and inconsistent. Is there any supporting reference for this claim?
18. Line1043-1081:These two parts can be integrated together for the discussion.
19. There are too many format errors in the references and the inconsistent formats. Please check and adjust it.
20. Lines 1082-1087, the conclusion emphasizes the relevance of UAVs in the application potential and solutions to constraints in smallholder farmlands. However, the discussion regarding the study of NUS spatial distribution is insufficient and deviates slightly from the research objective stated in the abstract.
21. Is the distinction between cited articles containing hypotheses, popular opinions, and undisputed facts clearly made?
22. Please check if the naming sequence of each image in the article is accurate. Figure 3 and Figure 4 are missing from the image numbering, and image numbers are missing in line 487 and other lines such as line 495.
23. Figure 1 and Figure 17 are similar, and it is possible to consider merging them into one figure.
The sentence expression is chaotic, and many words in the sentence are inappropriate. The writing of the article is terrible.
Reviewer 2 Report
It is an article that is within the scope of the journal. I brought an interesting topic: NUS. The problem is well contextualized and relevant to the SDGs' debate.
The methodology (PRISMA) is adequate.
The work has a section (Results) that is not well organized. It deserves attention.
The discussion has some points to be added (suggestions), but it does not detract from the established discussion, as well as the presented conclusions.
When reading the manuscript, some passages brought me doubts. I make my considerations for the appreciation of the authors.
Opening paragraphs need revision. The sentences are wide, making it difficult for the text to flow. A restructuring of the order (the development of the idea) will be enough.
I draw attention to acronyms, such as NUS. There is no explanation in the introduction. The word already appears directly on line 69 and two examples of plants on line 71. I suggest a brief explanation where the meaning and main characteristics of the plants that comprise this category are addressed. I suggest that this definition is also in the summary.
As much as it is a specialized and international journal, the communication must also provide the (lay) reader with conditions to understand the terms presented. For example, the initials RS, SSA and ML appear without saying what they mean. Due to the facilities of the translation mechanisms, it would help even more to give precision to the translated phrase.
When saying that he was going to deal with sweet potatoes and taro in particular, he failed to mention which spatial area. I understand that these plants may be relevant in the context of South Africa, but we cannot deduce this for Argentina or Somalia.
Results section.
In general, graphics/images can be better worked/edited. A lot of important information is “hidden”. The work carried out is important and deserves to be better explored. Results. Suggestion: Improved layout and data presentation.
The figures found on lines 301 and 302 have a lot of information. The layout of it doesn't help and understand the flow. I suggest editing the figures, choosing the flow with the highest intensity and the lowest. All information at once does not allow for proper reading.
Evaluate swapping some column charts for a table. As organized as it is tiresome, it does not help explain the rationale for analyzing NUS in the southern hemisphere or even if these are being studied in South Africa.
Some specific points to be appreciated.
1) Improve data on Underutilized Crop Species. This category has a huge scope of plants. In this sense, it would be interesting to define in a table which plants make up Underutilized Crop Species and which sensors were identified or analyzed. Figures 7 and 9 can be grouped together.
Question: Which NUS/sensor had the highest frequency by region. Graphs and names in quotation marks do not help to systematize the reasoning. Figure 9A is an example that could be distributed by region.
2) Due to the temporal issue, these were not evaluated. Questions:
How was the theme approached during the analysis?
To what extent has the use of remote sensing identified areas of NUS?
Which institutions evaluated the issue the most in the time frame?
Could NUS be associated with any temporal trend?
If these questions (1 and 2) are answered, it will help clarify a doubt I had when reading the manuscript: Wouldn't the argument about global trends in the South be underestimated? I believe this point should be addressed in the discussion session.
Reviewer 3 Report
Dear AuthorsI am glad to review this review article. According to my openion the authors should need to modify some sections before final publications.
You can find some comments below
1. The author(s) should the abstract section. Abstract section is complex and moreover abstract section doesnot resflect the whole study.
2. Line 52: Add space between text and reference.
3. I am not satisfied with materials and methods section. Its not look like a reiew article.
4. The author(s) should read more literaure about the UAVs and then discuss about the methodology and material in section 2.
5. Line 220: What do you mean by meanwhile. Is there something is missing?
6. Results are good but the author (s) should focus on the material and methods sections.
7. Check all figures and mark caption a, b and c and then dicuss each figure in the results section.